



# Seasonal forecasting of snow resources at Alpine sites

Silvia Terzago[1], Giulio Bongiovanni[1,2], and Jost von Hardenberg[3,1]

[1]Institute of Atmospheric Sciences and Climate, National Research Council, Torino, Italy
[2]Department of Civil, Environmental and Mechanical Engineering, University of Trento, Trento, Italy
[3]Department of Environment, Land and Infrastructure Engineering, Politecnico di Torino, Torino, Italy

**Correspondence:** Silvia Terzago (s.terzago@isac.cnr.it)

**Abstract.** Climate warming in mountain regions is resulting in glacier shrinking, seasonal snow cover reduction, changes in the amount and seasonality of meltwater runoff, with consequences on water availability. Droughts are expected to become more severe in the future with economical and environmental losses both locally and downstream. Effective adaptation strategies involve multiple time scales, and seasonal forecasts can help in the optimization of the available snow/water resources with

lead times of several months. We developed a prototype to generate seasonal forecasts of snow depth and snow water equivalent with starting date November $1^{st}$ and lead times of 7 months, so up to May $31^{st}$ of the following year. The prototype has been co-designed with end users in the field of water management, hydropower production and of mountain ski tourism, meeting their needs in terms of indicators, time resolution of the forecasts, visualization of the forecast outputs. In this paper we present the modelling chain, based on the seasonal forecasts of ECMWF and Météo-France seasonal prediction systems, made available

through the Copernicus Climate Change Service (C3S) Climate Data Store. Seasonal forecasts of precipitation, near-surface air temperature, radiative fluxes, wind and relative humidity are bias-corrected and downscaled to three sites in the Western Italian Alps, and finally used as input for the physically-based multi-layer snow model SNOWPACK. The RainFARM stochastic downscaling procedure is applied to precipitation data in order to allow an estimate of uncertainties due to the downscaling method.

The skill of the prototype in predicting the monthly snow depth evolution from November to May in each season of the hindcast period 1995-2015 are demonstrated using both deterministic and probabilistic metrics. Forecast skills are determined with respect to a simple forecasting method based on the climatology, and station measurements are used as reference data. The prototype shows very good skills at predicting the tercile category, i.e. snow depth below- and above-normal, in the winter (lead time 2-3-4 months) and spring (lead times 5-6-7 months) ahead: snow depth is predicted with higher accuracy (Brier

Skill Score) and higher discrimination (Area Under the ROC Curve Skill Score) with respect to a simple forecasting method based on the climatology. Ensemble mean monthly snow depth forecasts are significantly correlated with observations not only at short lead time 1 and 2 months (November and December) but also at lead time 5 and 6 months (March and April) when employing the ECMWFS5 forcing. Moreover the prototype shows skill at predicting extremely dry seasons, i.e. seasons with snow depth below the $10^{th}$ percentile, while skills at predicting snow depth above the $90^{th}$ percentile are model-, station- and

score-dependent. No remarkable differences are found among the skill scores when the precipitation input is bias-corrected, downscaled or bias-corrected and downscaled compared to the case in which raw data are employed, suggesting that skill scores are weakly sensitive to the treatment of the precipitation input.



## 1 Introduction

Mountain snowpack provides a natural reservoir which supplies water in the warm season for a variety of uses, such as hydropower production and irrigated agriculture in and downstream of mountain areas. However warming trends often amplified in mountain regions (Pepin et al., 2015; Palazzi et al., 2019) have resulted in glacier shrinking, seasonal snow cover reduction and changes in the amount and seasonality of runoff in snow dominated and glacier-fed river basins (Pörtner et al., 2019). Future cryosphere changes are projected to affect water resources and their uses (Pörtner et al., 2019). Current warm winter seasons may become normal at the end of the $21^{st}$ century, and there is indication for droughts to become more severe in the future (Haslinger et al., 2014; Stephan et al., 2021; Stahl et al., 2016). Effective adaptation strategies to address and reduce future water scarcity involve multiple time scales, from the seasonal scale, for the optimization of the available water resources with few months lead time, to climate scales, for the long-term planning of water storage infrastructures and the diversification of mountain tourism activities (Calì Quaglia et al., 2021). In these wide range of time scales, seasonal predictions have been considered with growing interest for their potential to provide early warning of extreme seasons, and to enable decision makers to take necessary actions to minimize negative impacts.

The ability of the current seasonal forecasts systems at predicting the main meteorological variables (air temperature and precipitation) is generally limited in the extra-tropics (Mishra et al., 2019) and this is reflected on poor streamflow prediction (Greuell et al., 2018; Arnal et al., 2018; Wanders et al., 2019; Santos et al., 2021). Some skill is found for the winter season streamflow prediction in about 40% of the European domain (Arnal et al., 2018), while contrasting results are found for high altitude catchments, where the discharge is mostly related to snow and ice melt. Some studies highlighted better skill than surrounding areas (Anghileri et al., 2016; Santos et al., 2021), while others found poor streamflow predictions due to the lack of snowpack predictability in the Alpine region (Wanders et al., 2019). One of the issues in mountain streamflow forecasting is the lack of reliable information to initialize physically based streamflow models, for example in terms of distribution of snow water equivalent (SWE) and of soil moisture, and this often results in limited forecasting skill (Li et al., 2019). In addition to initialization issues, multi-model ensemble streamflow predictions generally employ hydrological models in which the representation of snow processes is simplified and snow accumulation and melt are poorly captured (Wanders et al., 2019). These studies highlight the importance of a reliable representation of mountain snowpack for improving streamflow forecasts in mountain areas. An original approach to seasonal hydrological forecasting in mountain areas is to change the focus from the prediction of instantaneous hydrological fluxes (rainfall, streamflow) to that of slowly varying, and probably more predictable, hydrological quantities, such as the snow water equivalent (Förster et al., 2018). Snowpack is a natural "integrator" of the climatic conditions over multiple days/months, so even if daily temperature and precipitation forecasts do not match the corresponding observations, the differences may compensate over monthly/seasonal time scales and provide reasonable monthly/seasonal snowpack forecasts. Several economic activities recognized a value in seasonal forecasts of mountain snow accumulation, either per se or as an indicator of the meltwater available in the season ahead: i) public water managers, who can prepare strategies to mitigate the negative effects of extremely dry or extremely wet seasons, ii) hydropower companies involved in reservoir management, who use forecasts of the snowpack evolution to decide whether to release or save water in



the reservoir; iii) mountain ski resorts managers, for which seasonal snowpack predictions are relevant to estimate the amount of artificial snow to be produced (Marke et al., 2015).

The seasonal predictability of snow-related variables has so far been rarely studied. Kapnick et al. (2018) explored the
potential of predicting the snowpack in March with 8 months lead time (starting date July $1^{st}$) in the western US, using three atmosphere-ocean general circulation models (AOGCM) at different resolutions (200, 50 and 25 km). That study showed a good correlation to observations in most parts of the area, demonstrating the feasibility of such kinds of forecasts. However, regions with high hydroclimate variability, i.e. maritime mountain ranges, still represent a challenge for the predictability at seasonal time scale. Bellaire et al. (2011) coupled the 1D snow cover model SNOWPACK with the Canadian weather forecasting model
GEM15 to simulate snow depth at Mount Fidelity (Canada) for the winter season, from October to May, 2009-2010. They found that the agreement between simulated and measured snow depth depends on the precipitation filtering method, stressing the importance of the precipitation input on the snowpack simulations. In the Alpine region, Förster et al. (2018) tested a method to derive deterministic predictions of the sign of February SWE anomalies, i.e. SWE below- or above-average, over the Inn headwaters catchment. They set up a rather simple framework in which a distributed water balance model driven by
seasonal forecasts of monthly air temperature and precipitation anomalies provides SWE anomaly forecasts over the basin. This forecasting method showed some skill in predicting the sign of the basin-average SWE anomaly and, more in general, it proved the higher robustness of SWE predictions compared to precipitation ones. However the deterministic approach adopted in this study does not allow to obtain a quantification of the uncertainty of the forecasts, and the only information on the sign of the SWE anomaly without an associated probability of occurrence is of limited usefulness in practical applications. In
complex modeling chains the accuracy of the output variables is subject to multiple sources of uncertainty, which are present in the various components of the modelling chain: the meteorological forecast system(s) employed; bias adjustment eventually applied to adjust systematic errors in the models; downscaling techniques eventually applied to mitigate the mismatch between the scale of the forcing and the scale at which snow processes occurs; the process model employed, its setup and initialization. Each component of the chain should be evaluated to assess its relative contribution to the overall forecasting error, however
this analysis is often overlooked or not adequately performed.

In this study we present a method to generate for the first time multi-model multi-member seasonal forecasts of mountain snow depth/water equivalent during the period from November to May of the following year, taking advantage of the state-of-the-art modelling techniques. We developed a prototype which uses seasonal forecasts of the main meteorological variables available produced by multi-system seasonal forecast service of the Copernicus Climate Change Service (C3S) to simulate
the snowpack evolution at a given mountain site. Seasonal forecast system outputs at 1°x1° spatial resolution and daily or 6-hourly temporal resolution are bias-corrected and downscaled using different techniques depending on the variable type and characteristics (i.e. instantaneous or flux variable) to generate km-scale, hourly forcings. This fine scale hourly forcing is employed to drive the physical, multi-layer, 1-dimensional snow model SNOWPACK (Lehning et al., 2002) which proved to be one of the best performing models in a recent benchmark study (Terzago et al., 2020). The prototype is run at each location,
and at each location it provides ensembles of snow depth seasonal forecasts at hourly time step, which are then aggregated to monthly or seasonal scale for the analysis.



The prototype is demonstrated at three selected sites in the Western Italian Alps, where snow seasonal forecasts can be exploited by stakeholders in the fields of hydropower energy production, water management and ski resort management. Ensemble seasonal forecasts are evaluated using both deterministic and probabilistic metrics (Wilks, 2011) to assess different

forecast features (accuracy, discrimination and sharpness) at the monthly and seasonal scales. The skill of the forecast system is assessed compared to a reference forecast based on the past observations at in-situ stations. We also present an evaluation of the uncertainty associated with each step of the modelling chain, for example verifying the impact of using meteorological inputs from different seasonal forecast systems, and alternatively applying bias-adjustment or downscaling methods or a combination of both.

The paper is organized as follows. Section 2 describes the study area, the data used, the modelling chain and the forecasting skill assessment methods. Section 3 presents the prototype results, followed by a discussion and final conclusions in Sect. 4 and 5, respectively.

## 2 Methodology

The prototype has been co-designed with stakeholders, who provided guidance on the features required to make this climate

service useful for applications. Although the purpose of the prototype is to respond to specific needs of the users, it has been developed to be general, flexible and applicable to any area of study for which seasonal snow forecasts are needed. In the following we present the motivations for the study, that closely determine the area of evaluation of the prototype, the datasets employed and a step-by step description of the methodology.

### 2.1 Motivation for the work, domain of study and in-situ data

The prototype has been conceived for applications in the Western Italian Alps, in three Valleys which are relevant for different stakeholders (Fig. 1), i.e. i) the Orco Valley, hosting an artificial water reservoir serving a plant for hydropower production; ii) the Ala Valley, relevant for water supply to the Metropolitan City of Torino, 2.2 million inhabitants; and iii) the Upper Sesia Valley, which hosts one of the largest ski resorts in the Western Italy, at the foot of Monte Rosa. All stakeholders are interested in seasonal forecasts of snow abundance to plan in advance activities and investments for the season ahead. In particular they are

interested in forecasting low snow seasons to limit snow/water shortage and economic losses. Each area of study hosts at least one station which provides nivo-meteorological data since the 1990s useful to evaluate model outputs. For each station, name, geographical position, variables provided and start/end of the station activity are reported in Table 1. All stations are situated at elevations above 2000 m a.s.l. and at these altitudes a critical variable to measure is total precipitation, which is typically underestimated by standard (unheated) pluviometers. A quality check of the station data showed that increases in snow depth

are often associated with daily total precipitation equal or close to zero. This suggests that standard pluviometers strongly underestimate solid precipitation, so total precipitation measurements are considered unreliable during the snow season and they have not been used in the analysis.




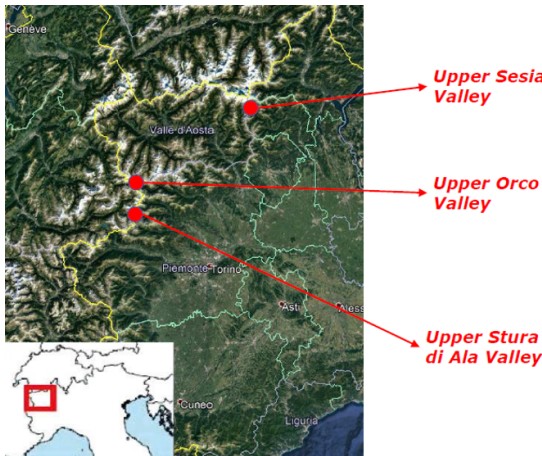

**Figure 1.** Map of the study sites indicating the three nivo-meteorological stations in NW Italian Alps (©Google Maps 2021).

**Table 1.** Stations considered in this study, elevation, position and date of start of automatic meteorological station records.

|  | Station | | |
|---|---|---|---|
|  | **Bocchetta delle Pisse** | **Lago Agnel** | **Rifugio Gastaldi** |
| Valley | Sesia | Orco | Stura di Ala |
| Elevation (m a.s.l.) | 2410 | 2304 | 2659 |
| Latitude (WGS 84, °) | 45.875556 | 45.467778 | 45.298056 |
| Longitude (WGS 84. °) | 7.901111 | 7.139167 | 7.143333 |
| Air temperature | 01/01/1988 | 11/10/1996 | 30/04/1988 |
| Total precipitation | 06/07/1996 | 12/10/1996 | 05/07/1996 |
| Wind Speed | 01/01/1990 | - | 01/01/1990 |
| Total radiation | 22/03/2012 | - | 06/10/2017 |
| Snow depth | 01/01/1995 | 01/11/1997 | 01/01/1995 |
| Fresh snow depth | 01/01/1995 | 01/11/1997 | 01/01/1995 |

## 2.2 ERA5 reanalysis

In addition to observational data we use the latest ECMWF global reanalysis product, ERA5 (Hersbach et al., 2020), which
provides reanalysis fields at 0.25° (about 30 km) spatial resolution and 1 hour temporal resolution. Compared to the previous
reanalysis product, ERA-Interim, ERA5 uses one of the most recent versions of the Earth system model and data assimilation
methods applied at ECMWF and modern parameterizations of Earth processes. With respect to ERA-Interim, ERA5 also has
an improved global hydrological and mass balance, reduced biases in precipitation, and refinements of the variability and trends





**Table 2.** C3S seasonal forecast systems used in this study and relative characteristics in terms of Institution running the simulations, ensemble size and main reference.

| Model | Institution | Ensemble size | Reference |
|---|---|---|---|
| ECMWFS5 | European Centre for Medium-Range Weather Forecasts (ECMWF) | 25 | Johnson et al, 2019 |
| MFS6 | Météo-France | 25 | Dorel et al., 2017 |

of surface air temperature (Hersbach et al., 2020). To supply the lack of trusted observational data, we use the ERA5 reanalysis

at the gridpoint closest to each station to run a reference simulation with the snow model. This simulation is the benchmark against which to evaluate the seasonal snow depth forecasts.

### 2.3 Seasonal forecast data

We employ historical forecasts (hindcasts) from ECMWF System 5 (ECMWFS5) and Météo-France System 6 (MFS6) models (Table 2) obtained from the Copernicus Climate Data Store (https://climate.copernicus.eu/). We considere hindcasts initialized

each November $1^{st}$ and run for the 7 months ahead (November-May) over the period 1995-2015 (21 hindcasts) for which evaluation data (snow depth observations) were available for all the stations. We consider all the variables needed to force the snow model: 2m temperature, 2m dewpoint temperature, total precipitation, surface solar and thermal radiation downwards, soil temperature level 1, 10-meter U and V wind components. Original C3S flux variables (precipitation and radiation) are accumulated since the beginning of the forecast, so they have been converted to daily values (see Table 3 for details). Horizontal

wind components are converted to wind speed (modulus). Possible discrepancies between the climatologies of seasonal forecast and reference data (from observations, where available, or ERA5) have been investigated and adjusted using suitable methods as described in the following sections. Seasonal forecasts resolution is 1°Lon x 1°Lat in space and daily or 6 hourly in time. These resolutions are insufficient to simulate snow processes at the local scale, so we apply simple downscaling techniques to generate data at 1 km and 1 hour resolution. The applied techniques are specific for each variable and they are briefly described

in the following.

#### 2.3.1 Air temperature

Figure 2a shows the multi-year average of the November-May 2m air temperature from ECMWFS5 hindcasts compared to observations. The ECMWFS5 temperature bias is large and time-dependent, and the same happens for MFS6 seasonal forecast system (not shown). We test two different methods to correct this bias. In one case, we perform a quantile mapping correction

with respect to ERA5 2m temperature upscaled to the model resolution (1°) on a monthly basis (Fig. 2a, "QM Data"), then we adjust the average November-May bias of the "quantile-mapped data" with respect to observations (Fig. 2a, "Downscaled data"). In the second case, we directly correct the bias with respect to observations at the subdaily scale. In detail, we derive the multi-year average daily temperature bias of the original seasonal forecast data with respect to observations, then we





**Table 3.** C3S seasonal forecast model variables used to create the forcing for the prototype: original variable name, short name and units, variable short name and units after post-processing (see Sect. 2.3 for details).

| C3S variable | Short name | Units | Frequency | Short name CV* | Units CV* |
|---|---|---|---|---|---|
| 2m temperature | t2m | K | 6 h instantaneous | tas | K |
| 2m dew point temperature | d2m | K | 6 h instantaneous | tdps | K |
| Total precipitation | tp | m | 24h aggregation** | prlr | mm/day |
| Surface solar radiation downwards | ssrd | J/m$^2$ | 24h aggregation** | rsds | W/m2 |
| Surface thermal radiation downwards | strd | J/m$^2$ | 24h aggregation** | rlds | W/m2 |
| Soil temperature level 1 | tsl1 | K | 6 h instantaneous | tsl1 | K |
| 10 metre U and V wind components | u10, v10 | m/s | 6 h instantaneous | sfcWind | m/s |

*CV=Converted variable

**=since beginning of forecast

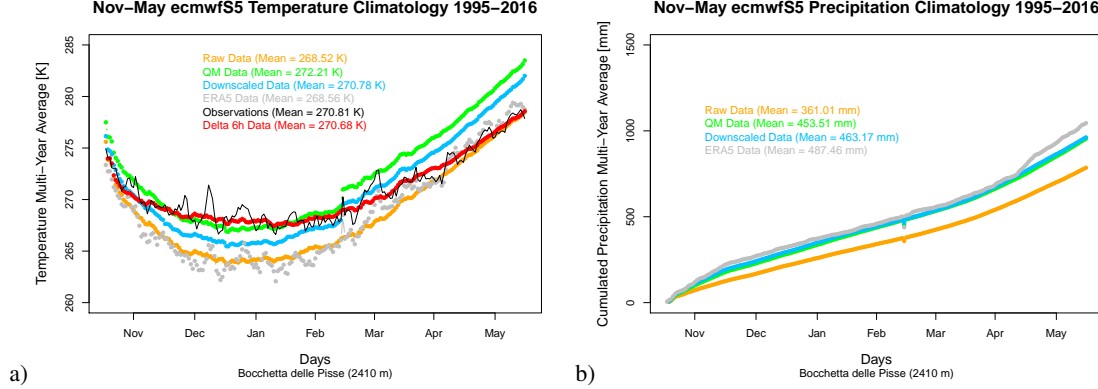

**Figure 2.** ECMWFS5 multi-annual (1995-2016) average of a) daily air temperature b) accumulated total precipitation at Bocchetta delle Pisse station (2410 m a.s.l.) in the ERA5 reanalysis (gray) and in seasonal forecast data with different levels of post-processing: (orange) raw data; (green) after the application of the quantile mapping at monthly scale; (cyan) after the application of the quantile mapping at monthly scale and the correction of the seasonal average bias; (red) after the correction of the 6-hourly mean temperature bias. For temperature we report the observed climatology derived from the in-situ station (black); this information is not reported for total precipitation since data are affected by large underestimation errors due to snow undercatch.

interpolate the bias in time to 6 hours and we finally apply this offset to the original model data (Fig. 2a "Delta 6h data").
While the first method is able to only partially correct the forecast climatology, leaving substantial underestimation in winter and overestimation in spring and producing jumps in the seasonal cycle due to the application of the quantile mapping at the monthly scale, the second method successfully reproduces the observed temperature seasonal cycle (by construction). In light of these results we adopt the second method to adjust temperature seasonal forecast bias.



### 2.3.2   Total precipitation

Figure 2b shows the discrepancy between the ECMWFS5 daily precipitation climatologies and the ERA5 reference: the bias has been adjusted with a rather sophisticated approach which allows to take into account orographic effects. First, daily precipitation seasonal forecasts have been adjusted by applying a quantile mapping approach (Gudmundsson et al., 2012; Perez-Zanon et al., 2021) on a monthly basis, using ERA5 total precipitation data upscaled to $1°$ as a reference dataset. Then bias-adjusted daily data have been downscaled from $1°$ to about 1 km using the RainFARM stochastic precipitation downscaling method

(Rebora et al., 2006; D'Onofrio et al., 2014) improved to take into account orographic effects (Terzago et al., 2018). This method employs orographic weights derived from a fine-scale precipitation climatology (WorldClim, Fick and Hijmans, 2017) to correct the downscaled field (Terzago et al., 2018). The RainFARM method is used to generate an ensemble of 10 stochastic realizations of the downscaled precipitation for each of the 25 seasonal forecast system ensemble members. This procedure allows generating 250-member ensemble forecasts for each starting date. Looking at the results in Fig. 2b, the quantile mapping

ping allows to accurately reconstruct the long term climatology of the accumulated precipitation, and this feature is conserved after the application of the RainFARM downscaling. After the application of the spatial downscaling, precipitation is then disaggregated in time, from daily to hourly resolution, by equally redistributing the precipitation amount over all time steps with sufficient relative humidity to allow precipitation. We chose RH>80% as a threshold.

### 2.3.3   Surface shortwave and longwave radiation downwards

Daily accumulated surface shortwave and longwave radiation downwards ($J/m^2$) have been converted into average daily radiation fluxes ($W/m^2$) and downscaled in space using a simple bilinear interpolation to the coordinates of the station using the Climate Data Operator command line tools (CDO, Schulzweida, 2019). In order to disaggregate in time average daily fluxes into hourly fluxes we employed a sort of analogue method using ERA5 as a reference. The choice of ERA5 as a reference dataset is supported by a high temporal correlation with both seasonal forecast systems, for each station and for both short-

wave (r>0.82) and longwave (r>0.51) radiation. For each day of the forecast period i) we consider the seasonal forecast of (shortwave, longwave) daily radiation for that day, ii) we consider all ERA5 daily average radiation values for that month over the period 1993-2019, iii) we sort the ERA5 daily values in ascending order, from the lowest to the highest, iii) we consider the 11 ERA5 values closest to the forecast for that day, iv) we randomly choose one among the 11 ERA5 daily values and we consider the corresponding 24 hourly values, v) we assume these 24 ERA5 hourly values to be the seasonal forecast of hourly

radiation for that day. This technique allows us to reconstruct hourly forecasts which are plausible for the specific month and which conserve the daily mean radiation forecast for that specific day.

### 2.3.4   Humidity, surface wind and soil temperature

Seasonal forecast models in the CDS archive do not provide directly specific or relative humidity among their output variables. So we derive relative humidity from air temperature and dew point temperature following Lawrence (2005). Air temperature,





dew point temperature as well as wind speed and soil temperature have been bilinearly interpolated to the coordinates of the station.

## 2.4   The SNOWPACK model

We simulate snow dynamics with the SNOWPACK model, a sophisticated snow and land-surface model allowing for a detailed description of the mass and energy exchange between the snow, the atmosphere and optionally the vegetation cover and the

soil (Bartelt and Lehning, 2002). It provides a detailed description of snow properties, including weak layer characterization, phase changes and water transport in snow (Hirashima et al., 2010). A particular feature is the treatment of soil and snow as a continuum with a choice of a few up to several hundred layers (Bartelt and Lehning, 2002). The model is able to accurately estimate mountain snow depth in a variety of meteorological conditions, with an average error of about 10 cm when forced by accurate in-situ data (Terzago et al., 2020). The SNOWPACK model is used in its default configurations, so no tuning of the

model parameters is made to locally improve the snow depth simulations.

In our framework the SNOWPACK model has to be initialized with measured snow depth on November $1^{st}$. We can have two different conditions: i) snow free soil or shallow or spatially discontinuous snow cover, or ii) continuous snow cover. We use a threshold of 10 cm to decide whether the initial snow cover is absent/shallow/discontinuous or not. If snow cover is lower than 10 cm SNOWPACK is initialized with snow depth equal to zero. Otherwise, SNOWPACK is initialized with the observed

snow depth and a snow profile which characterizes each snow layer. Since the snow profile is unavailable from observations we simulate it by running SNOWPACK over the previous summer and driving the model with a mix of reanalysis and observational data: all meteorological forcing are provided by ERA5 except for air temperature which is derived by observations. Simulations generally start on August $1^{st}$, or the following first day with observed snow depth SD=0, and end on $1^{st}$ November, providing the snow profile for that day, which is then used to initialize the SNOWPACK simulation in forecast mode.

## 2.5   Experiments with the SNOWPACK model

Precipitation is a critical parameter both to measure and to represent in model simulations. As explained in Sect. 2.3.2 we employ quite sophisticated techniques to bias-adjust and downscale precipitation forecasts to the station scale. Such complexity could be a limit in an operational framework where simple, easy-to-use approaches are preferred. To this aim we investigate a range of methods to correct precipitation inputs to verify if simpler methods can provide comparable results with respect

to the most complex ones. We devised a set of 4 experiments with the SNOWPACK model, differing in the treatment of the precipitation input, with the aim of evaluating the model sensitivity to the accuracy of the precipitation input. The experiments are reported in Table 4 and briefly summarized here: 1) the first experiment (RAW) uses original seasonal forecast precipitation data without any further refinement; 2) the second experiment (QM) uses precipitation data bias-adjusted with the quantile mapping method; 3) the third experiment (RainFARM) uses seasonal forecast precipitation data stochastically downscaled to 1

km with the RainFARM method; 4) the last experiment (QM+RainFARM) uses both the quantile mapping and the RainFARM methods to bias-adjust and downscale precipitation forecasts. For each experiment and each seasonal forecast system listed in





**Table 4.** Plan of experiments with the SNOWPACK model. The meteorological forcing is generated using ECMWFS5 and MFS6 seasonal forecast systems outputs

| Experiment | Total precipitation | Output ensemble members |
|---|---|---|
| RAW | RAW | 25 |
| QM | Quantile Mapping reference ERA5 ($QM_{ERA5}$) | 25 |
| RainFARM | RainFARM | 250 |
| QM+RainFARM | $QM_{ERA5}$+RainFARM | 250 |

Table 2 we run the modelling chain on a set of 21 meteorological forecasts starting on November $1^{st}$ of each year in the period 1995-2015.

## 2.6 Output of the modelling chain

For each experiment of Table 4, the output of the modelling chain consists of an ensemble of hourly (or eventually daily) snow depth time series representing the seasonal forecasts for the three considered stations. The number of ensemble members is 25 in the RAW and QM experiments and 250 in the RainFARM and QM+RainFARM experiments, i.e. 10 RainFARM precipitation downscaling realizations for each of the 25 model ensemble members (Table 4). An example of ensemble snow depth seasonal forecast for the season 2006/2007 is reported in Fig. 3 and it will be discussed in Sect. 3. In order to perform

the statistical analysis of the set of snow depth hindcasts, the output of the modelling chain originally at hourly time step is aggregated at the daily, monthly and seasonal (DJF, MAM and November-May) scale to be compared with in-situ station measurements.

## 2.7 Evaluation metrics

Hourly snow depth seasonal forecasts are first aggregated to daily data and then to monthly and seasonal means over the periods
November-May (NM), December-February (DJF) and March-May (MAM). The seasonal means are computed by using all corresponding daily data. Monthly and seasonal means are then evaluated by employing both deterministic and probabilistic metrics. While deterministic metrics consider the ensemble mean of the forecasts compared to the observations, probabilistic metrics compare different features of the forecast distribution with respect to the observations or the observed distribution. In the following we briefly describe all the metrics used in this study:

– Time correlation: The simplest way to evaluate ensemble forecasts is to assess the time correlation between ensemble mean forecasts and observations. Since we are interested in assessing the correlation of fluctuations, the linear trend in time series has been removed and the correlation has been calculated on residuals. The correlation is expressed as Pearson's correlation coefficient, the confidence interval is computed by a Fisher transformation and the significance level relies on a one-sided student-T distribution, with threshold 0.95 (BSC-CNS et al., 2021)



– Brier Score (BS): Among the set of probabilistic scores the Brier Score represents the mean square error of the probability forecast for a binary event, e.g. snow depth in a given tercile of the distribution (Mason, 2004). In our analysis, continuous forecasts are first transformed into tercile-based forecasts (i.e. probabilities for snow depth forecast to fall into the lower, middle or upper tercile of the forecast distribution) as suggested in (Mason, 2018). Then, the BS is calculated for each tercile. We also explored the forecast skill in predicting extreme events, i.e. the BS associated to monthly and seasonal

snow depth below the $10^{th}$- and above the $90^{th}$-percentile of the forecast distribution. Tercile and percentile thresholds are calculated over the reference period 1995-2015

   – Area Under the ROC curve (AUC): The Receiver Operating Characteristic Curve (ROC, Jolliffe and Stephenson, 2012) similarly to the Brier Score, allows the evaluation of binary forecasts. Given an ensemble forecast for a binary event, for example snow depth in the upper tercile, the ROC curve shows the true-positive rate against the false-positive rate

for different probability threshold settings. The area under the ROC curve, shows the ability of the forecast system to discriminate between "event" and "non-event", i.e. it is a measure of the discrimination of the forecast system. AUCs are calculated separately for each tercile and then averaged over the three terciles

   – Continuous Ranked Probability Score (CRPS): One of the most widely used accuracy metrics for ensemble forecasts is the Continuous Ranked Probability Score Matheson and Winkler (1976). The CRPS is the integrated squared difference

between the forecast cumulative distribution function (CDF) and the empirical (observed) CDF, which is a step function. The CRPS has a negative orientation, i.e. the lower the score the better the forecast CDF approximates the observed CDF. The perfect value for CRPS is 0.

   In this study the BS, AUC and CRPS scores are presented as skill scores (SS). The skill scores directly indicate the skill of the forecast system with respect to a reference forecast, in our case the climatological forecast, derived from the set of

climatological values except for the value that occurred. Values of the skill scores are negative when the forecast system is worse than the climatological forecast and positive when the forecast system is better than the climatological forecast. A skill score of 0 indicates no improvements with respect to the climatological forecast. BSS and CRPS are calculated for each starting date and lead time, then averaged over all starting dates and converted into skill scores as follows:

$$SS = \frac{S - S_{ref}}{S_{perf} - S_{ref}} \tag{1}$$

where SS is the value of the skill score, S is the value of the score of the forecast system against the observations, $S_{ref}$ is the value of the score of the climatological forecast against the observations and Sperf is the value of the score in the theoretical case that forecasts perfectly match observations. The AUC Skill Score (AUCSS), instead, is derived using the following formula (Wilks, 2011):

$$AUCSS = 2(AUC - 0.5) \tag{2}$$





The uncertainty on the time correlation and the skill scores has been evaluated by estimating the confidence interval (CI) using the bootstrap method (Bradley et al., 2008; Wilks, 2011), as recommended by Mason (2018). Bootstrapping is widely used to find the sampling distribution of a quantity and then to compute its standard error and CI. At first, given n the number of ensemble members, depending on whether n is odd or even, $n/2$ or $(n+1)/2$ members are randomly selected with replacement. Thus, a skill score is computed considering only selected ensemble members. The procedure has been iterated 1000 times

generating a sample distribution, from which mean and 90% confidence interval error bars are estimated.

## 3    Results

### 3.1    An example of snow depth forecast

Figure 3 represents an example of snow depth forecast for the season 2006/2007 referring to the station of Bocchetta delle Pisse. The forecast is derived using the meteorological forcing provided by the ECMWFS5 model, post-processed as described

in Sect. 2.3. Precipitation forecasts have been bias-adjusted with the quantile mapping method and then downscaled to 1 km with the RainFARM method (QM+RainFARM experiment) generating 10 stochastic realizations for each of the 25 forecast ensemble member (250 downscaled precipitation forecasts in total). The ensemble spread, the 5-95$^{th}$ percentile range and the ensemble median of the forecasts for the season 2006/2007 are compared to the ensemble median of all forecasts for all seasons of the period 1995-2015 in order to highlight the characteristics of the considered season with respect to model climatology

and determine if snow depth is expected to be below or above median. The plot also reports the snow depth observations for that season and the observed climatology to visually inspect the accuracy of the forecast (please note that differences between the observed and the modelled climatology are due to uncertainties in the bias-adjusted meteorological forcing and in the snow model structure).

We present the output of the modelling chain also in the form of tercile-based forecasts (Figure 4). For each month of the

season, the tercile-based forecast plot shows the probability density function (PDF) of the 250 monthly mean snow depth forecasts, together with the probabilities to have snow depth in each tercile, and the indication of the most likely tercile. The plot also reports the probability for snow depth to be lower than the 10$^{th}$ percentile and higher than the 90$^{th}$ percentile. Tercile and percentile thresholds are calculated on the 21*250 monthly mean snow depth forecast values over the period 1995-2015. In the example reported in Figure 4 snow depth forecasts indicate the lower tercile (below normal) as most likely in each month

of the snow season. In order to visually evaluate the quality of the forecast, the observed snow depth is also reported: if the observed snow depth falls within the most likely tercile, the forecast is successful. In this season the forecast is successful in February, March, April and May, so in late winter and spring.

### 3.2    Effects of the precipitation bias-adjustment and downscaling

The forecast presented in Figs. 3 and 4 is obtained after applying quite sophisticated bias correction and downscaling techniques

to precipitation data. In this section we assess the added value, if any, of applying those bias-adjustment and/or downscaling

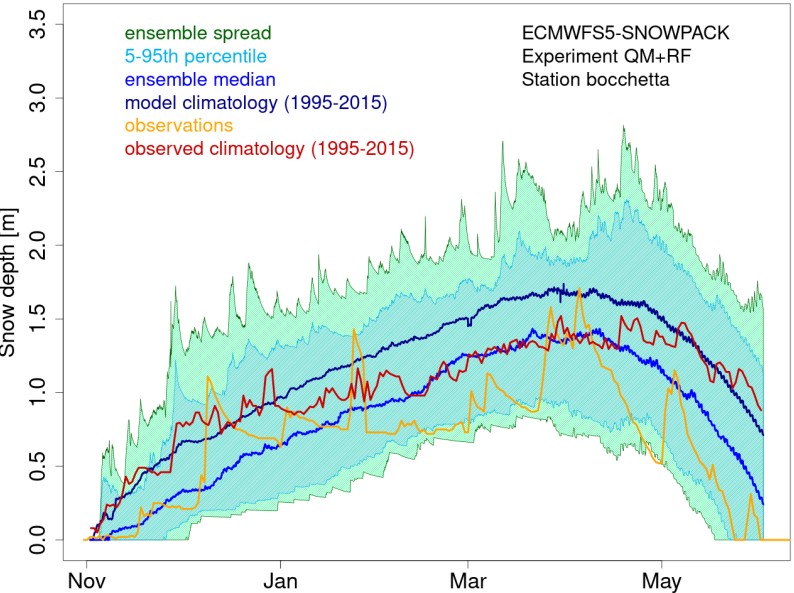

**Figure 3.** ECMWFS5-SNOWPACK snow depth ensemble forecasts (QM+RainFARM experiment, 250 ensemble members) initialized on November $1^{st}$ 2006 and issued for the 7 months ahead, for the site of Bocchetta delle Pisse (2410 m a.s.l., North Western Italian Alps). Dark green lines represent the ensemble spread, cyan lines represent the $5^{th}$-$95^{th}$ percentile range of the snow depth distribution, the blue line represents the ensemble median of the snow forecasts over the considered season, the dark blue line represents the ensemble median of the forecasts over the reference period 1995-2015, the orange line represents in-situ observations and the red line represents the median of the observations over the reference period 1995-2015.

methods compared to the use of raw precipitation data. We present the results of the 4 experiments (RAW, QM, RainFARM, QM+RainFARM) listed in Table 4, in which we apply or not the correction methods to precipitation forecasts. We use an indirect approach, i.e. we assess the added value of total precipitation corrections by measuring the agreement between the snow depth climatology obtained from the 4 experiments and observed climatology in terms of root mean square error (RMSE).

For each of the two forecast systems, ECMWS5 and MFS6, and each experiment, Figure 5 shows the simulated snow depth climatology (multi-annual and multi-member average) compared to the observed climatology at the station of Bocchetta delle Pisse for the period 1995-2015. The corresponding RMSE is reported in Table 5.

When driven by ERA5 forcing, SNOWPACK RMSE on snow depth is in the range 0.30-0.35 m for Bocchetta delle Pisse and Lago Agnel stations, while it is higher (RMSE=0.5 m) for Rifugio Gastaldi: in this last station, snowfalls are typically

followed by rapid snow ablation (not shown), so the large RMSE can be related to ERA5 issues in capturing the meteorological conditions responsible for the fast melting. ERA5-driven simulations are the reference against which to compare seasonal-forecast-driven simulations. Compared to the ERA5-driven run, the RAW experiment shows remarkably lower RMSE when using the ECMWFS5 forcing and remarkably higher RMSE when using the MFS6 forcing. This suggests that after the bias-





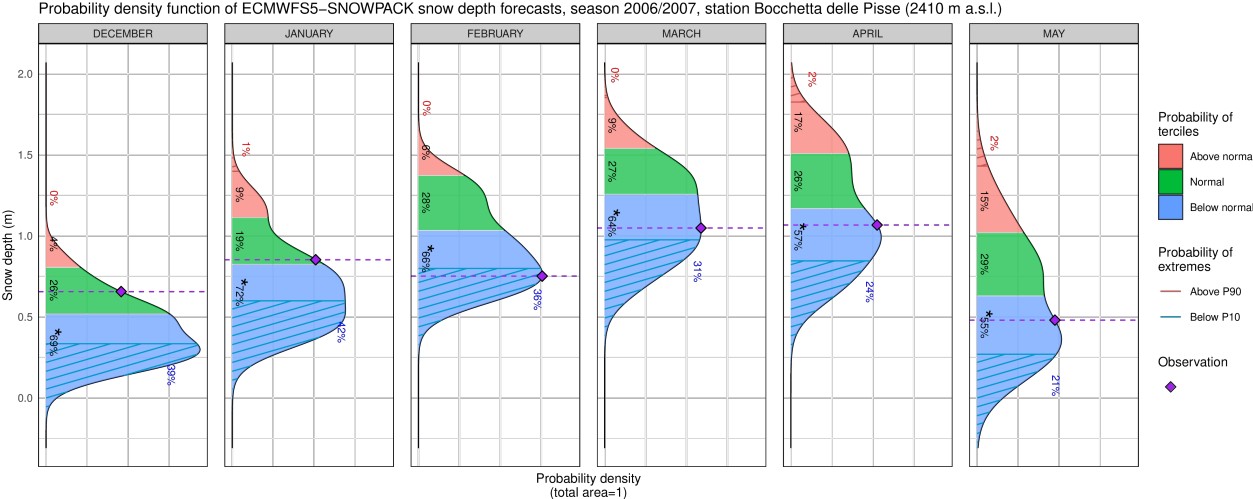

**Figure 4.** Probability Density Functions (PDFs) of the ECMWFS5-SNOWPACK monthly mean snow depth ensemble forecasts for the season 2006-2007 and for the station of Bocchetta delle Pisse, 2410 m a.s.l. in the Italian Alps. Areas in coral, green and blue colors represent the % probability to have monthly average snow depth below, near and above the normal conditions for the period, respectively, and the asterisk indicates the most likely tercile. Areas with blue and red parallel lines represent the probability to have monthly snow depth below the $10^{th}$ percentile and above the $90^{th}$ percentile, respectively. Observations are reported as purple diamonds.

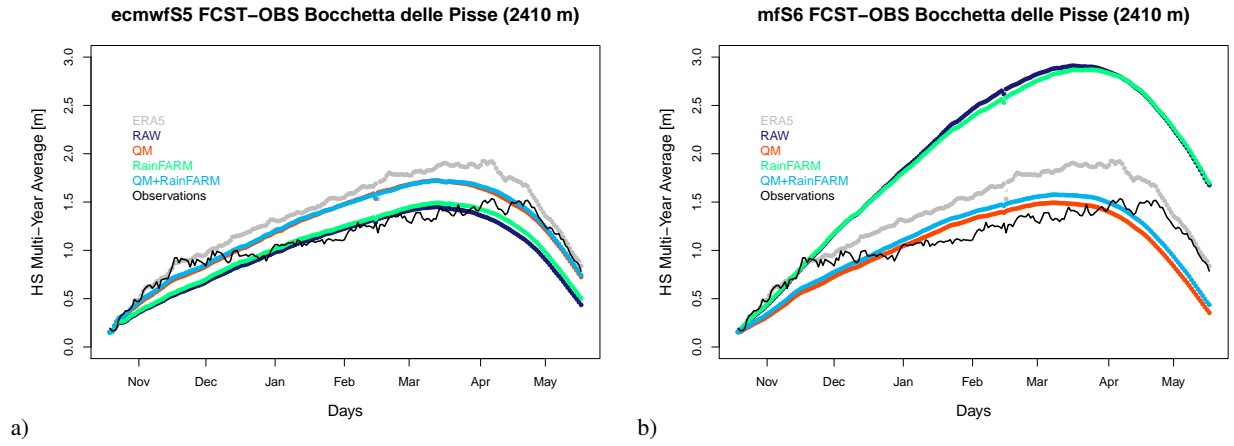

**Figure 5.** Daily snow depth climatology for the period 1995-2005 as simulated by the SNOWPACK model forced by ERA5 (gray) and by (a) ECMWFS5 and (b) MFS6 seasonal forecasts data with different precipitation input (RAW, QM, RainFARM and QM+RainFARM) as specified in Table 4, for the site of Bocchetta delle Pisse. Observations are reported in black for comparison.





**Table 5.** RMSE between simulated and observed daily snow depth climatologies at the station of Bocchetta delle Pisse for the experiments listed in Table 4. Model simulations are obtained by forcing SNOWPACK with ERA5, ECMWFS5 and MFS6 meteorological variables. ECMWFS5 and MFS6-driven experiments (RAW, QM, RF and QM+RF) differ in the treatment of total precipitation (see Table 4).

| | RMSE [m] | | | | | | | | |
| | ERA5 | ECMWFS5 | | | | MFS6 | | | |
| | | RAW | QM | RF | QM+RF | RAW | QM | RF | QM+RF |
|---|---|---|---|---|---|---|---|---|---|
| Bocchetta delle Pisse | 0.31 | 0.19 | 0.21 | 0.16 | 0.21 | 1.04 | 0.21 | 1.01 | 0.20 |
| Rifugio Gastaldi | 0.50 | 0.27 | 0.35 | 0.30 | 0.45 | 1.38 | 0.39 | 2.13 | 0.57 |
| Lago Agnel | 0.32 | 0.18 | 0.22 | 0.37 | 0.60 | 1.19 | 0.24 | 2.63 | 0.78 |

adjustment of temperature seasonal forecasts i) the ECMWFS5 forcing is more accurate than the ERA5 reanalysis; ii) the
MFS6 forcing has residual systematic errors that affect the reliability of the simulations.

The application of the quantile mapping to heavily biased precipitation forecasts (MFS6) allows for a clear improvement of the model RMSE which is reduced up to almost 5 times compared to the RAW experiment. On the other hand, the application of the quantile mapping to already accurate forcing (ECMWFS5) can have different effects depending on the accuracy of the reference dataset. Here the application of the quantile mapping using ERA5 as a reference has the effect of slightly increasing the RMSE (see Table 5, ECMWFS5 model, QM experiment) but it might also have detrimental effects when the reference dataset is inaccurate.

The application of the RainFARM downscaling (RF experiment) produced small effects at Bocchetta delle Pisse station (orographic weight equal to 1.05), and gradually more relevant effects at Rifugio Gastaldi and Lago Agnel (weights equal to 1.21 and 1.43, respectively, see Terzago et al. (2018) for details). In these last two cases the orographic downscaling amplifies precipitation amounts and leads to an overestimation of the snow depth output, with snow depth errors doubling for about 50% increase in the precipitation input.

These results suggest that the choice of the forecast system strongly impacts the agreement between the simulated and the observed climatology. The application of the quantile mapping is recommended in case of large biases in the precipitation input, in order to reproduce a snow depth climatology as realistic as possible. However, the application of the quantile mapping is recommended only if a trusted, reliable reference dataset is available. In fact, if the reference dataset is less accurate than the dataset that we want to correct, the application of the bias adjustment may lead to larger errors. The RainFARM downscaling is blind to model biases so, in presence of heavily biased forcing, it should be applied only after bias correction. Since the downscaling might have either positive or negative effects depending on the orographic weights, the added value of the downscaling should be checked against observations before using the fine scale precipitation data.



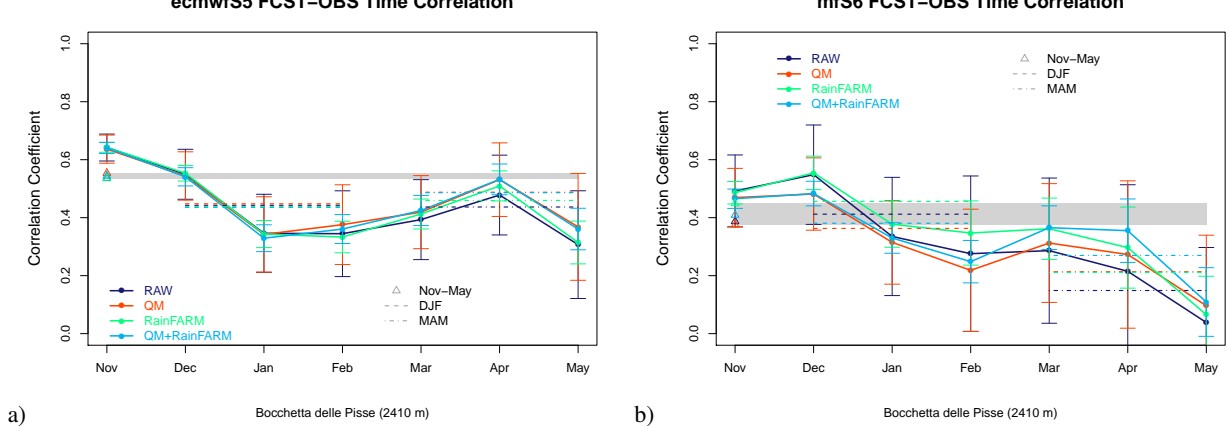

**Figure 6.** Pearson's correlation coefficient between forecasts of ensemble-mean monthly-mean snow depth obtained with (a) ECMWFS5 and (b) MFS6 forcing and observations at the site of Bocchetta delle Pisse. Forecasts are initialized on November $1^{st}$ and run with a lead time of 7 months. Colored dots represent the correlation for each month and each experiment; horizontal dashed (dash-dotted) lines represent DJF (MAM) values; the gray rectangle and the 4 colored triangles represent seasonally-averaged (Nov-May) values. Error bars represent the 5-95$^{th}$ percentile range of the distribution of 1000 bootstrap samples as described in Sect. 2.7.

### 3.3 Evaluation of the snow depth forecasts

In order to assess the skill of the forecasting method presented in this study we evaluate the snow depth forecasts over the period 1995-2015 (hindcasts) in comparison to snow depth observations, using the set of metrics introduced in Sect. 2.7. We recall that all metrics are calculated on detrended time series.

#### 3.3.1 Time correlation

Figure 6 shows the correlation between ensemble mean monthly and seasonal hindcasts and observations for the two seasonal forecast systems, ECMWFS5 and MFS6, and for the four experiments listed in Table 4 for the station of Bocchetta delle Pisse. Confidence intervals represented in Fig. 6 as error bars or as a gray rectangle correspond to the 5-95$^{th}$ percentile range of 1000 bootstrap samples derived as described in Sect. 2.7. The correlation values for all three stations, together with their significance at 95% confidence level, are reported in Table 6.

A common behavior is found among all stations: the correlation is highest in November, i.e. at lead time 1-month when the meteorological input is generally well correlated with observations, then the correlation decreases reaching a minimum in winter months (January or February depending on the station and forcing). After February the correlation increases to a secondary maximum in April, then it finally drops in May. Correlation values are very similar among different experiments, especially for the ECMWFS5 model. The largest differences among experiments are found for the MFS6 model in spring





**Table 6.** Time-correlation of the detrended mean monthly snow depth forecasts with respect to observations at the three stations for ECMWFS5 and MFS6 systems. Correlations significant at 95% confidence level are identified in bold and by an asterisk (*).

| | | Pearson time correlation | | | | | | | | | | | | | |
| | | BOCCHETTA DELLE PISSE | | | | | RIFUGIO GASTALDI | | | | | LAGO AGNEL | | | | |
| | | ECMWFS5 | | | | | | | | | | | | | | |
| | ERA5 | RAW | QM | RF | QM+RF | ERA5 | RAW | QM | RF | QM+RF | ERA5 | RAW | QM | RF | QM+RF |
|---|---|---|---|---|---|---|---|---|---|---|---|---|---|---|---|
| Nov | 0.93* | 0.64* | 0.64* | 0.64* | 0.64* | 0.44* | 0.74* | 0.76* | 0.73* | 0.75* | 0.91* | 0.62* | 0.66* | 0.65* | 0.68* |
| Dec | 0.92* | 0.55* | 0.54* | 0.55* | 0.54* | 0.52* | 0.59* | 0.59* | 0.59* | 0.60* | 0.89* | 0.51* | 0.53* | 0.53* | 0.53* |
| Jan | 0.88* | 0.35 | 0.34 | 0.34 | 0.33 | 0.71* | 0.46* | 0.45* | 0.43* | 0.45* | 0.84* | 0.37 | 0.38 | 0.39* | 0.39* |
| Feb | 0.84* | 0.34 | 0.38* | 0.33 | 0.36 | 0.71* | 0.37* | 0.38* | 0.36 | 0.39* | 0.89* | 0.35 | 0.37 | 0.41* | 0.43* |
| Mar | 0.85* | 0.39* | 0.42* | 0.41* | 0.42* | 0.67* | 0.41* | 0.39* | 0.39* | 0.40* | 0.89* | 0.43* | 0.41* | 0.50* | 0.48* |
| Apr | 0.79* | 0.48* | 0.53* | 0.51* | 0.53* | 0.60* | 0.40* | 0.39* | 0.41* | 0.38* | 0.92* | 0.43* | 0.44* | 0.49* | 0.46* |
| May | 0.80* | 0.31 | 0.37 | 0.31 | 0.36 | 0.66* | 0.30 | 0.30 | 0.30 | 0.30 | 0.93* | 0.26 | 0.29 | 0.33 | 0.32 |
| Seas | 0.89* | 0.54* | 0.55* | 0.54* | 0.54* | 0.66* | 0.52* | 0.53* | 0.52* | 0.53* | 0.92* | 0.48* | 0.51* | 0.54* | 0.54* |
| DJF | 0.90* | 0.44* | 0.45* | 0.44* | 0.44* | 0.65* | 0.50* | 0.50* | 0.48* | 0.51* | 0.88* | 0.42* | 0.45* | 0.46* | 0.47* |
| MAM | 0.82* | 0.44* | 0.49* | 0.46* | 0.49* | 0.70* | 0.39* | 0.39* | 0.39* | 0.39* | 0.93* | 0.41* | 0.41* | 0.46* | 0.44* |
| | | MFS6 | | | | | | | | | | | | | | |
| | ERA5 | RAW | QM | RF | QM+RF | ERA5 | RAW | QM | RF | QM+RF | ERA5 | RAW | QM | RF | QM+RF |
| Nov | 0.93* | 0.49* | 0.47* | 0.49* | 0.47* | 0.44* | 0.44* | 0.45* | 0.43* | 0.44* | 0.91* | 0.30 | 0.28 | 0.29 | 0.29 |
| Dec | 0.92* | 0.55* | 0.48* | 0.55* | 0.48* | 0.52* | 0.46* | 0.43* | 0.45* | 0.42* | 0.89* | 0.45* | 0.40* | 0.45* | 0.41* |
| Jan | 0.88* | 0.34 | 0.31 | 0.38* | 0.33 | 0.71* | 0.21 | 0.29 | 0.16 | 0.24 | 0.84* | 0.23 | 0.27 | 0.20 | 0.25 |
| Feb | 0.84* | 0.28 | 0.22 | 0.35 | 0.25 | 0.71* | 0.11 | 0.16 | 0.09 | 0.12 | 0.89* | 0.23 | 0.27 | 0.21 | 0.27 |
| Mar | 0.85* | 0.29 | 0.31 | 0.36 | 0.37 | 0.67* | 0.15 | 0.24 | 0.12 | 0.20 | 0.89* | 0.29 | 0.41* | 0.24 | 0.36 |
| Apr | 0.79* | 0.21 | 0.27 | 0.30 | 0.36 | 0.60* | 0.09 | 0.17 | 0.06 | 0.13 | 0.92* | 0.19 | 0.30 | 0.18 | 0.28 |
| May | 0.80* | 0.04 | 0.10 | 0.07 | 0.11 | 0.66* | 0.03 | 0.07 | 0.01 | 0.05 | 0.93* | 0.09 | 0.14 | 0.10 | 0.16 |
| Seas | 0.89* | 0.39* | 0.38* | 0.45* | 0.41* | 0.66* | 0.22 | 0.29 | 0.18 | 0.25 | 0.92* | 0.28 | 0.32 | 0.25 | 0.33 |
| DJF | 0.90* | 0.41* | 0.36 | 0.46* | 0.38* | 0.65* | 0.27 | 0.31 | 0.23 | 0.28 | 0.88* | 0.31 | 0.33 | 0.29 | 0.33 |
| MAM | 0.82* | 0.15 | 0.21 | 0.21 | 0.27 | 0.70* | 0.06 | 0.14 | 0.04 | 0.10 | 0.93* | 0.16 | 0.26 | 0.15 | 0.24 |

(March and April), when QM and QM+RainFARM experiments provide higher time correlations than the RAW experiment, although they lie within the uncertainty range of the RAW experiment and none of these correlations is statistically significant.

Focusing on significant correlations at 95% confidence level (Table 6), we observe differences between seasonal forecast systems: using ECMWFS5 forcing, correlations are significant for all stations, all experiments and most lead times: the correlation is significant at lead time 1- and 2-month (November and December, respectively) and, interestingly, also at lead time

5- and 6-months (March and April), at the seasonal (November-May), winter (DJF) and spring (MAM) scale. Correlation is generally not statistically significant in May, and for some stations (Bocchetta delle Pisse and Lago Agnel) and experiments also in January and February. Compared to ECMWFS5, MFS6 correlation is considerably lower and generally not statistically significant after December, probably owing to a lower skill and larger biases in the meteorological forcing.





In challenging conditions such as poor meteorological forcing (MFS6) the application of bias-adjustment, downscaling or
the combination of both, generally improves correlations with respect to the RAW experiment, however this improvement does
not lead to statistically significant correlations.

### 3.3.2 Brier Skill Score

The Brier Skill Score (BSS) shows the relative skill of the forecast prototype with respect to the climatological forecast in terms
of mean square error of the probability forecasts for a binary event. In our case the binary event is "snow depth in a given tercile
of the forecast distribution". BSS takes positive values whenever the forecast prototype is more skillful than climatology. Figure
7 shows the time evolution of BSS for the two seasonal forecast systems, ECMWFS5 and MFS6, and for the four experiments
listed in Table 4 for the station of Bocchetta delle Pisse. Error bars computation are based on 1000 bootstrap samples derived
as described in Sect. 2.7. The winter (DJF), spring (MAM) and seasonal (Nov-May) BSS values are reported in the plot as
dashed lines, dot-dashed lines, and grey strips respectively, and the values for all three stations are reported and compared in
Figure 8, where positive (negative) BSS are highlighted in greenish (reddish) colors.

The BSS is generally positive for both seasonal forecast systems, both lower and upper terciles, for almost all experiments,
all lead times and all stations (Figs. 7 and 8). The BSS is generally highest in November and/or December and then it decreases
reaching its minimum, but with still positive values, in May (Fig. 7), demonstrating a clear added value of the prototype
forecast with respect to the climatological forecast. ECMWFS5 generally shows higher BSS than MFS6 for both lower and
upper terciles, indicating better forecast skills than MFS6. MFS6 shows large differences between the four experiments, without
a clear relation between the prototype skill and the application of the bias-adjustment and downscaling methods to precipitation
data.

### 3.3.3 Area Under the ROC curve Skill Score

AUCSS is a measure of the "discrimination" of the seasonal forecast system: it indicates how good are individual hindcasts
at discriminating mean monthly snow depth falling in the upper, middle and lower tercile in comparison to the reference
climatological forecast. We recall that positive values indicate improvements, while negative values indicate poorer skills than
the reference climatological forecast. Figure 9 shows the time evolution of AUCSS for the two seasonal forecast systems,
ECMWFS5 and MFS6, for the four experiments listed in Table 4, for the station of Bocchetta delle Pisse and for the lower
and upper terciles. Error bars are calculated based on 1000 bootstrap samples derived as described in Sect. 2.7. The winter
(DJF) and spring (MAM) AUCSS values for all three stations are reported in Figure 10, where positive (negative) AUCSS are
highlighted in greenish (reddish) colors.

Considering the ECMWFS5 forecasting system, a clear added value emerges in predicting the events in the terciles below
normal and above normal for all stations, all experiments and all lead times at least up to April included (up to May for Lago
Agnel, not shown). For all stations the AUC skill scores at the seasonal scale (DJF and MAM) indicate an improvement with
respect to the climatological forecast.



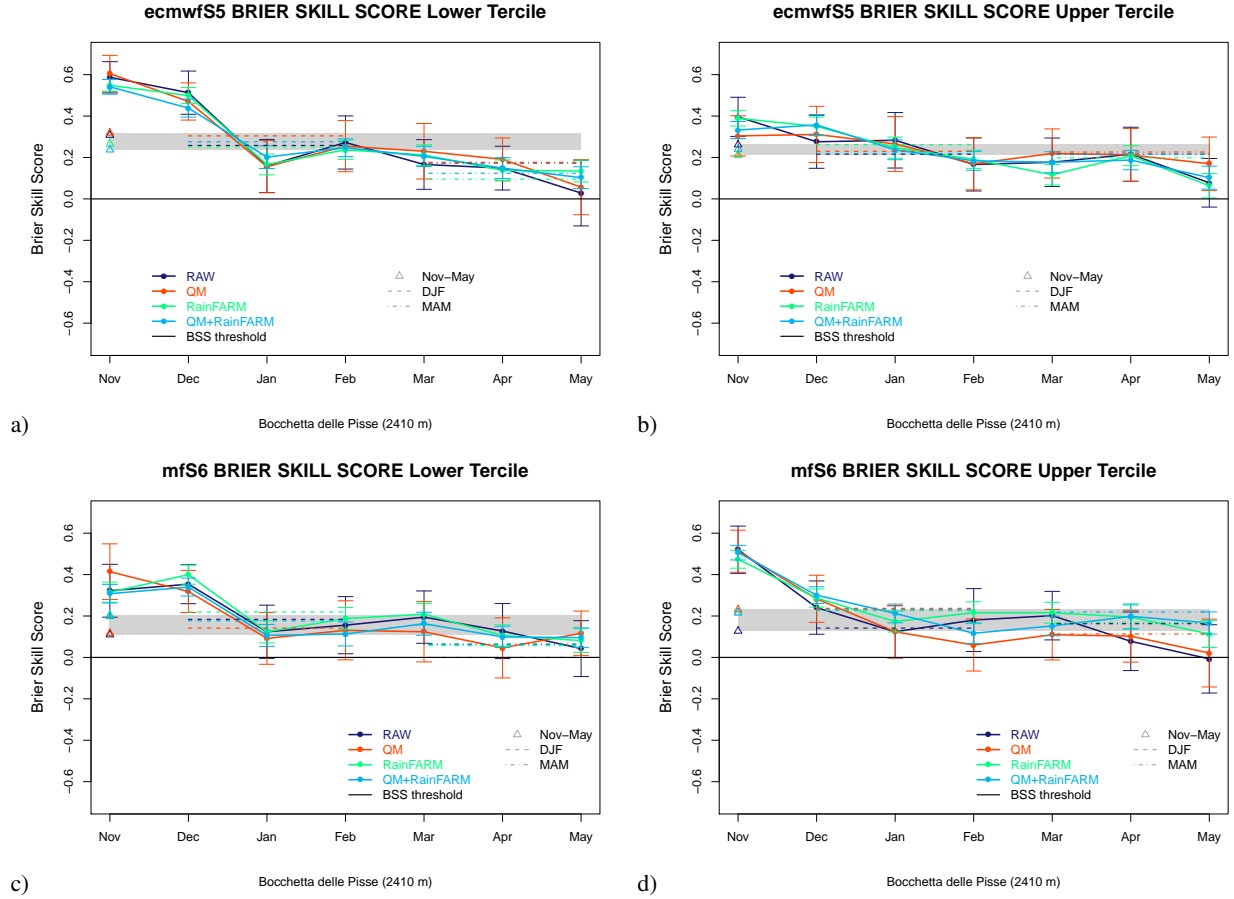

**Figure 7.** Brier Skill Score for seasonal forecasts of monthly- and seasonally-averaged snow depth in the (a,c) lower and (b,d) upper terciles, for (a,b) ECMWFS5 and (c,d) MFS6 forcing, starting date November $1^{st}$, lead times from 1 to 7 months for the site of Bocchetta delle Pisse. Colored dots represent the BSS for each month and each experiment; horizontal dashed (dash-dotted) lines represent DJF (MAM) BSS values; the gray filled rectangle and the 4 colored triangles all refer to the seasonal (Nov-May) values, indicating the BSS spread (min-max) and the single BSS values for the 4 experiments, respectively.





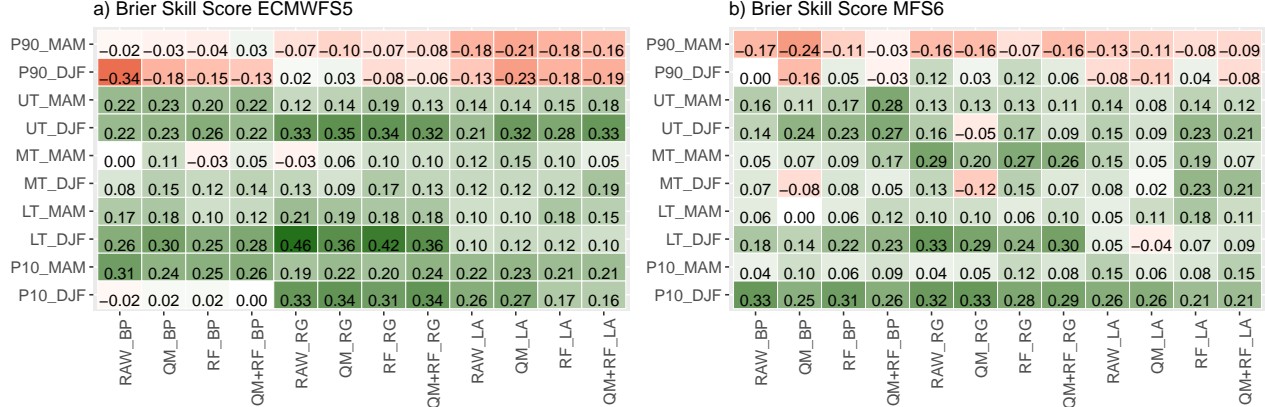

**Figure 8.** Brier Skill Score of the detrended seasonal (DJF, MAM) snow depth forecasts in the lower (LT), middle (MT), upper (UT) tercile, as well as in the lower (P10) and upper (P90) extreme of the distribution, with respect to the climatological forecasts, using observations at the three stations as a reference, for a) ECMWFS5 and b) MFS6 systems. Positive and negative BSSs are highlighted in greenish and reddish colors, respectively.

Considering the MFS6 forecast system, we find a clear added value at forecasting snow depth in the upper tercile (DJF and MAM) and in the lower tercile in DJF. The prediction skills for MAM snow depth in the lower tercile depend on the station: for example, good skills are found for Lago Agnel but not for Bocchetta delle Pisse.

Seasons with snow depth within the norm are predicted with similar skills as the climatological forecast, independently of
the forecast system, showing limited added value.

It is interesting to note that limited to the upper tercile, the AUCSS generally shows a secondary maximum in March or April (particularly evident for Rifugio Gastaldi and Lago Agnel stations, not shown) indicating that the forecast system has skills at predicting spring seasons with above normal snow depth. For the lower tercile this secondary maximum is generally not present otherwise, if present, it is generally not remarkable.

The largest differences among the four experiments are found for MFS6, however there is not a single experiment usually performing better than others.

### 3.3.4 Continuous ranked probability score (CRPS)

The continuous ranked probability score (CRPS) is a measure of the overall accuracy of the ensemble forecast. The Brier score and the CRPS are complementary measures, with the former providing information on the accuracy of tercile-based forecasts
and the latter evaluating the overall accuracy of the forecast distribution, considering the entire permissible range of values for the considered variable. Figure 11 shows the time evolution of CRPSS for the two seasonal forecast systems, ECMWFS5 and MFS6, and for the four experiments listed in Table 4 for the station of Bocchetta delle Pisse. In addition to the plots, Figure 12



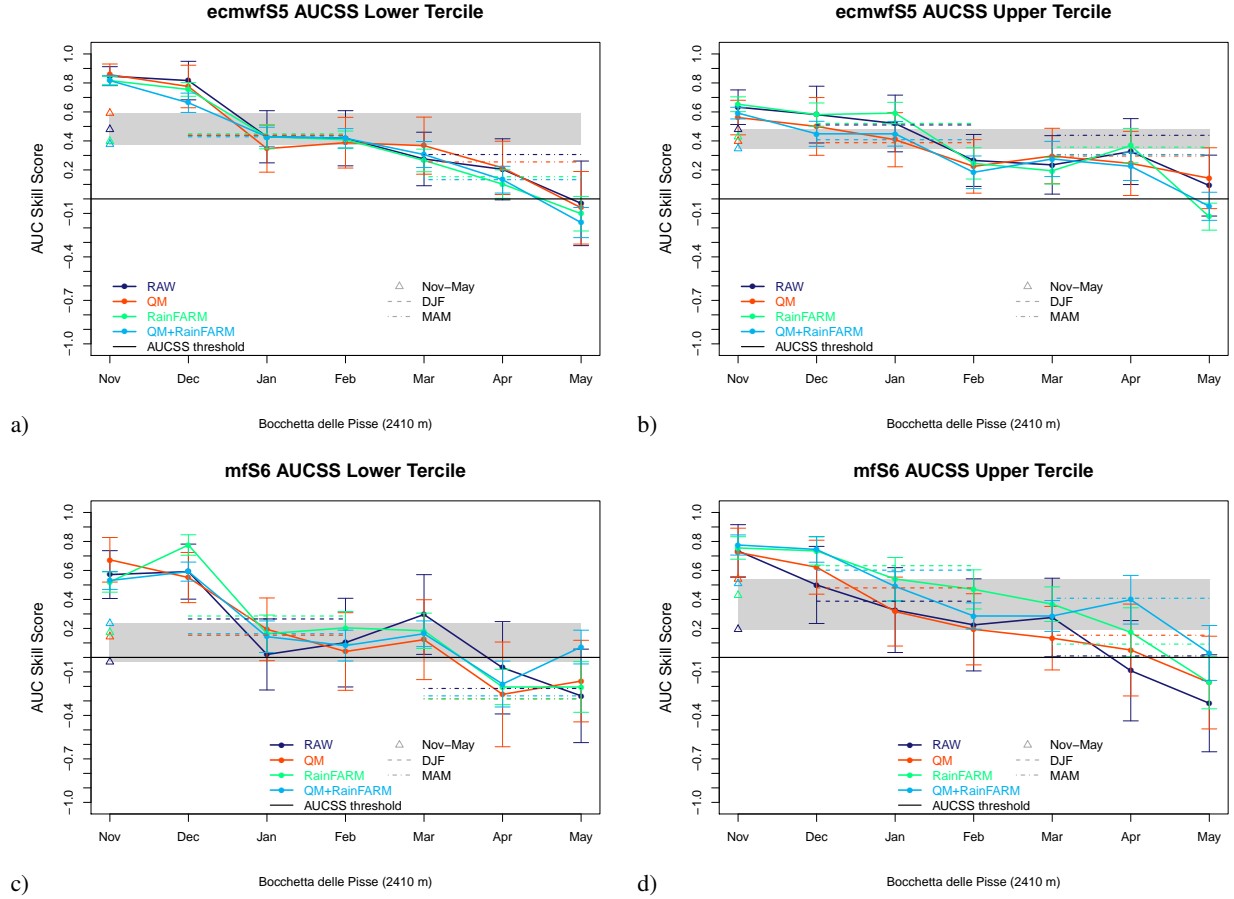

**Figure 9.** AUCSS for seasonal forecasts of monthly- and seasonally-averaged snow depth in the (a,c) lower and (b,d) upper terciles, for (a,b) ECMWFS5 and (c,d) MFS6 forcing, starting date November $1^{st}$, lead times from 1 to 7 months for the site of Bocchetta delle Pisse. Colored dots represent the AUCSSs for each month and each experiment; horizontal dashed (dash-dotted) lines represent DJF (MAM) AUCSS values; the gray filled rectangle and the 4 colored triangles all refer to the seasonal (Nov-May) snow depth forecasts, indicating the AUCSS spread (min-max) and the AUCSS values for the 4 experiments, respectively.





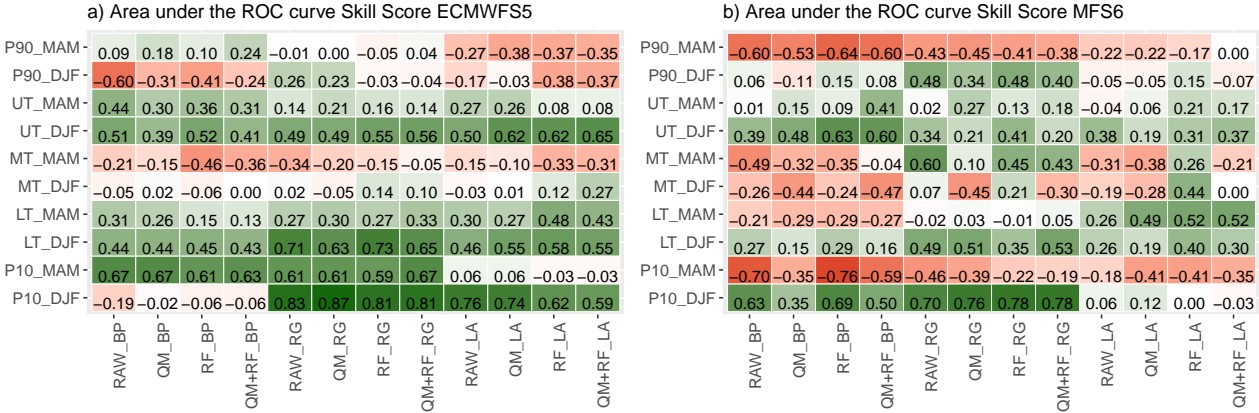

**Figure 10.** AUCSS of the detrended seasonal (DJF, MAM) snow depth forecasts in the lower (LT), middle (MT), upper (UT) tercile, as well as in the lower (P10) and upper (P90) extreme of the distribution, with respect to the climatological forecasts, using observations at the three stations as a reference, for a) ECMWFS5 and b) MFS6 systems. Positive and negative AUCSSs are highlighted in greenish and reddish colors, respectively.

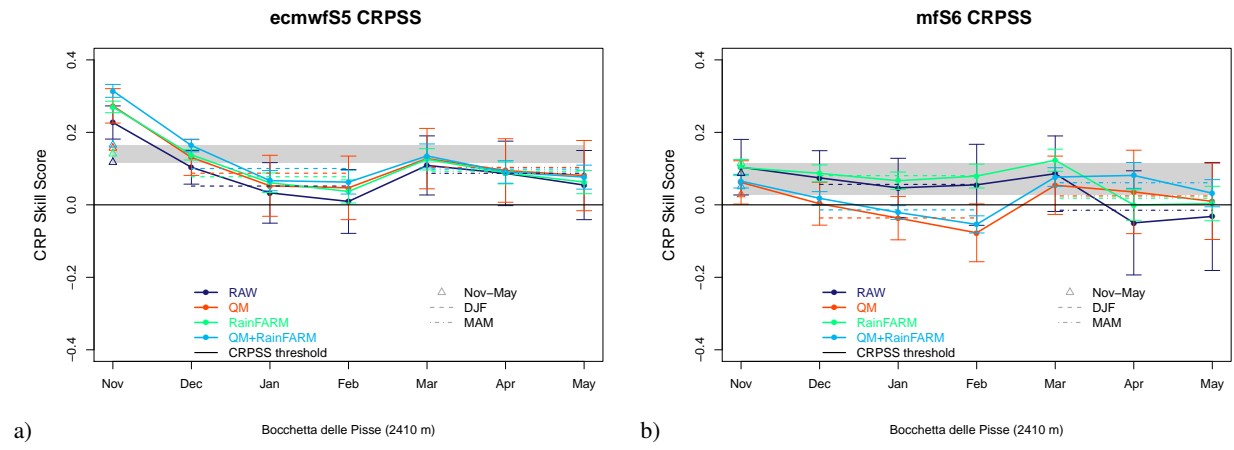

**Figure 11.** CRPSS for seasonal forecasts of monthly- and seasonally-averaged snow depth for (a) ECMWFS5 and (b) MFS6 forcing, starting date November $1^{st}$, lead times from 1 to 7 months, for the site of Bocchetta delle Pisse. Colored dots represent the scores for each month and each experiment; horizontal dashed (dash-dotted) lines represent DJF (MAM) scores; the gray filled rectangle and the 4 colored triangles all refer to the seasonal (Nov-May) snow depth forecasts, and they indicate the score spread among the 4 different experiments and the score for each of the 4 experiments, respectively.

shows the monthly and seasonal CRPSS values for all three stations, with values above 0.05 (below -0.05) are highlighted in green (orange).





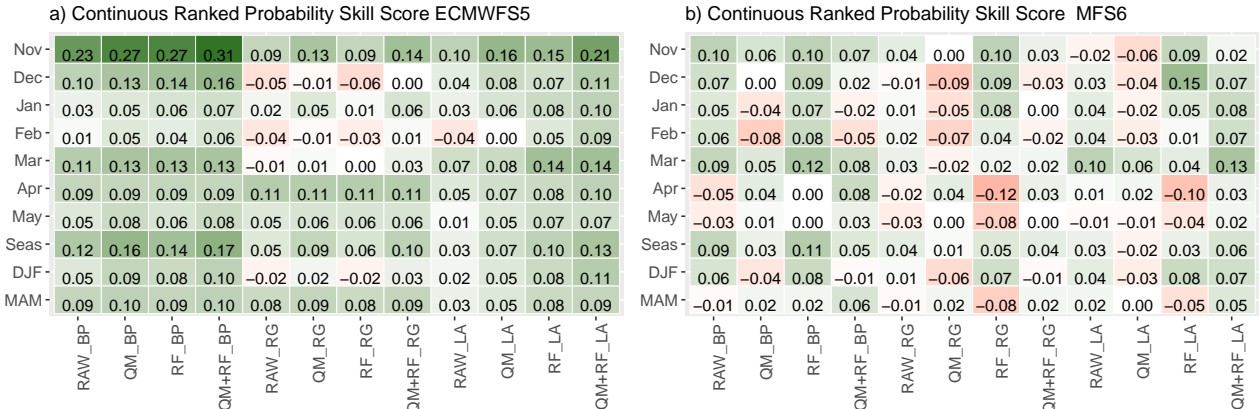

**Figure 12.** CRPSS of the detrended monthly and seasonal (DJF, MAM) snow depth forecasts with respect to the climatological forecasts, using observations at the three stations as a reference, for a) ECMWFS5 and b) MFS6 systems. Positive and negative CRPSSs are highlighted in greenish and reddish colors, respectively.

Considering the ECMWFS5 forecasting system, the CRPSS is generally positive, although with small values, across the different experiments, lead times and most of stations. Few exceptions with CRPSS values close to zero are found, and they are mostly in winter months. When using the MFS6 forecasting system, the skill is lower than ECMWFS5: the skill is present up to lead times 5-months (March) only for selected stations and experiments. Skills at lead time 6-7 months (April, May) and/or in the QM experiment are rare, suggesting a worsening of the performances when the total precipitation input is bias-adjusted with the quantile mapping method with respect to ERA5. The application of the quantile mapping with respect to ERA5 seems to waste some of the limited forecast skill.

Like many other skill scores analyzed, also the CRPSS decreases from November up to the end of the winter, then it increases again for a secondary maximum in March or April. This behavior is very common and it seems a robust feature across different forecast systems, experiments and test sites. Overall, the presence of positive CRPSS values, also when the score reaches its minimum, clearly indicates the added value of the prototype forecast than the climatological forecast, in terms of overall accuracy.

### 3.3.5 Events outside the 10-90$^{th}$ percentile range

The analysis of the prototype performance also covers the ability to predict events below the 10$^{th}$ percentile (P10, lower extreme) and above the 90th percentile (P90, upper extreme). Figure 13 shows the time evolution of BSS for extreme values for the two seasonal forecast systems, ECMWFS5 and MFS6, and for the four experiments listed in Table 4 for the station of Bocchetta delle Pisse. Figure 8 summarizes the BSS values for all the stations. Looking at the plots for Bocchetta delle Pisse station for the events below P10 (Figure 13a, 13c), the BSS is generally positive during the snow season, indicating a clear skill at predicting low snow months/seasons. In only one case the BSS is close to zero in all experiments (i.e. EMWFS5





forcing, DJF season, Bocchetta delle Pisse station) and the application of bias correction, downscaling or the combination

of both do not improve the skill. In all other cases, the skill is robust across different forecast systems, seasons, experiments and stations. It is interesting to note that MFS6 shows good skills at forecasting months/seasons with snow below P10, with similar performances or even outperforming the ECMWFS5-driven experiments. Looking at the plots for Bocchetta delle Pisse station for the events above P90 (Figure 13b, 13d), the BSS is generally negative, indicating no skill of the forecast system at predicting months/seasons with exceptionally abundant snow depth. This property is maintained considering different driving

models, seasons, experiments and stations. Some skill (positive BSS) are found for Rifugio Gastaldi station (all experiments) and especially when using the MFS6 forcing.

## 4  Discussion

In this paper we present an original prototype for generating multi-model ensemble seasonal forecasts of snow depth at the local scale from November up to May of the following year (7 months lead time), providing information which are relevant for

economic activities such as hydropower production, water management and winter ski tourism. The prototype is based on the SNOWPACK model forced by meteorological data of the Copernicus Climate Data Store seasonal forecast systems, namely ECMWFS5 and MFS6. The skill of the prototype has been assessed using different deterministic and probabilistic metrics: i) the time correlation of the ensemble mean snow depth forecast with the observed snow depth; ii) the accuracy (BSS) and the discrimination (AUCSS) of the tercile-based forecasts; iii) the accuracy of the forecast distribution (CRPSS). All probabilistic

skills have been calculated with respect to a simple forecast method based on the climatology (reference).

The prototype shows clear skill in tercile-based forecasts, i.e. higher accuracy (BSS) and higher discrimination (AUCSS) at forecasting events below and above normal compared to the climatological forecasts, independently of the driving seasonal forecast system, station, season and experiment considered. The prototype also shows skill at forecasting extreme snow seasons with snow depth below the $10^{th}$ percentile, while it has difficulties in predicting extremely snowy seasons (snow depth above

the $90^{th}$ percentile).

The choice of the forecast system has an impact on the skill of the prototype, with ECMWFS5 providing more robust skill across different seasons, metrics, and experiments than MFS6. The ECMWFS5-driven prototype provides high and significant time correlation between ensemble mean snow depth forecasts and observations for different time aggregations of the forecasts, i.e. over the whole period November-May, at the seasonal scale (DJF, MAM), or even at the monthly scale in November,

December, March April. These features are valid for all the three stations considered, and single stations provide even better results, with high and significant correlations also in January and February. By contrast, MFS6 shows significant correlation only at short lead times, i.e. November and/or December. The ECMWFS5-driven prototype shows skill at predicting the snow depth forecast distribution (CRPSS) at the November-May and MAM scale (all stations) and at DJF scale (for two out of three stations). On the contrary, MFS6 shows CRPSS values close to zero or slightly positive with a scattered pattern depending

on the station, season and experiment. In conclusion, compared to ECMWFS5, the MFS6 forcing prototype provides less widespread skills, and the performances are more score-, season-, experiment- and station-dependent.



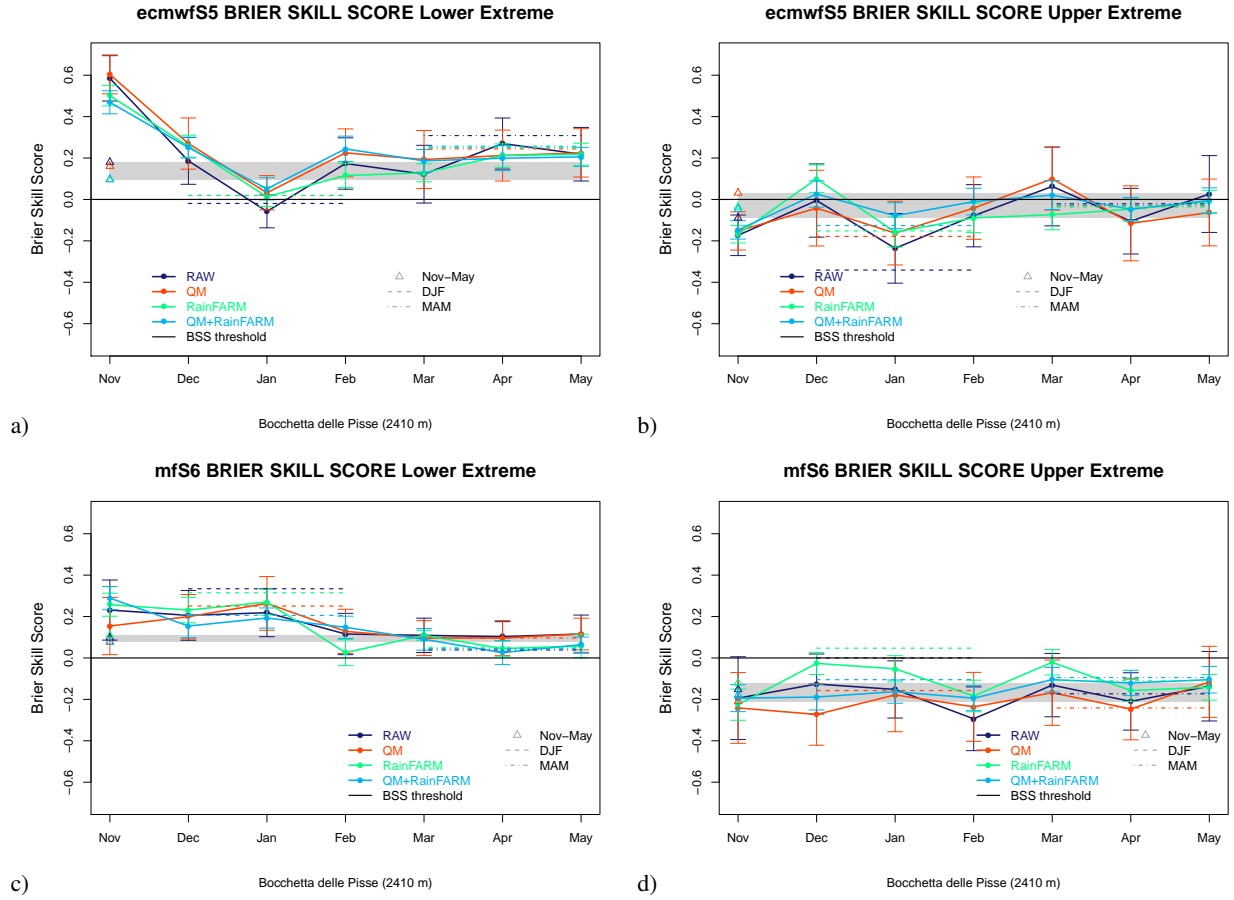

**Figure 13.** Brier Skill Score for seasonal forecasts of monthly- and seasonally-averaged snow depth (a,c) below the $10^{th}$ percentile (P10) and (b,d) above the $90^{th}$ percentile (P90), for (a,b) ECMWFS5 and (c,d) MFS6 forcing, starting date November $1^{st}$, lead times from 1 to 7 months, for the site of Bocchetta delle Pisse. Colored dots represent the BSS for each month and each experiment, horizontal dashed (dash-dotted) lines represent DJF (MAM) BSS values; the gray filled rectangle and the 4 colored triangles all refer to the seasonal (Nov-May) snow depth forecasts, and they precisely indicate the BSS spread among different experiments and the BSS values for each of the 4 experiments, respectively.



A common feature of both driving systems is their better skill at predicting above- or below-normal snow depth compared to near-normal snow depth. This issue has been found in several previous works (e.g. Calì Quaglia et al., 2021; Athanasiadis et al., 2017) and it has been explained with the difficulty at predicting small rather than large amplitude anomalies.

A second common feature of the two seasonal forecast systems is the time evolution of the monthly correlation: as expected it is maximum at the beginning of the season and then it decreases, however, surprisingly, it increases again to a secondary maximum in April (or March). This feature can be probably related to the fact that the spring snowpack is determined by the climatic conditions over the previous months, and even modest skill in the prediction of the main meteorological drivers (temperature and precipitation) at short lead time are reflected in skill at predicting snowpack at longer lead times. So even if

temperature and precipitation forecasts do not match the corresponding observations at the monthly scale, they can match at a longer (seasonal) scale and allow for surprisingly good predictability of the snow accumulation. Moreover, enhanced climate predictability in winter due to teleconnections such as the North Atlantic Oscillation (Lledó et al., 2020) may increase the skill in forecasting snowpack in the following spring. Increasing agreement from mid-winter to spring has been found not only for the time correlation but also for other skill scores, although in this last case the signal is not consistent throughout all forecast

systems, terciles and experiments.

    A third common feature of the two seasonal forecast systems is their skill at forecasting seasons with extremely low snow seasons, with snow depth below the $10^{th}$ percentile. This result is in line with previous studies on tercile- or quintile-based streamflow prediction (Santos et al., 2021; Wanders et al., 2019) where some reliability is achieved in the lower tercile, for high forecast probabilities. In contrast, for the upper tercile and even clearer for the middle tercile, no reliability is found. This

finding leads to two considerations. First, seasonal forecasts of snow depth appear more robust than streamflow forecasts, the former having skill in lower and upper terciles and in extreme events below P10, the latter having some limited skill in the lower tercile only. Second, it seems relatively easier to predict low-snow than high-snow seasons: this feature is of key importance since the most relevant feature requested by end-users to be available from the prototype is the capability of anticipating the occurrence of low snow seasons.

The accuracy of seasonal snow forecasts is subject to multiple sources of uncertainty, which are present in the various components of the production chain, that are: forecasts of the meteorological forcing, bias adjustment methods, downscaling techniques, snow model employed, model setup and initialization. Consequently, each component has to be evaluated to assess its relative contribution to the overall forecasting accuracy.

## 4.1    The impact of the choice of the seasonal forecast system

At the time when our snow depth forecast prototype was developed only two seasonal forecast systems provided all the variables necessary to drive the snow model, namely ECMWFS5 and MFS6, so we considered these two. Of course, additional seasonal forecasts systems should be analyzed as soon as data become available, investigating also the skill of the multi-model ensemble compared to the models taken individually. From our results based on ECMWFS5 and MFS6, the choice of the seasonal forecast system strongly impacts the skill of the prototype in terms of time correlation between forecasted

and observed snow depth, which is higher, significant and more widespread during the snow season in ECMWFS5 than in





MFS6. The choice of the seasonal forecast system also impacts the ability of the prototype to provide forecast distributions close to the observed ones (CRPSS). However, the choice of the forecast system does not substantially affect the ability of the system at providing skillful tercile-based forecasts (BSS and AUCSS). This finding suggests that even heavily biased seasonal forecast systems such as MFS6 over the study area can provide skillful tercile-based snow depth forecasts. In a recent
study (Calì Quaglia et al., 2021) assessed similar skill scores in ECMWFS5 and MFS6 when predicting DJF temperature and precipitation over the Mediterranean region, except for the anomaly correlation which seems slightly better in MFS6 than in ECMWFS5.

## 4.2 The impact of precipitation bias correction

Accurate temperature and precipitation data are essential for simulating snow processes since the former controls the phase of
precipitation and snow melt, and the latter controls snow accumulation. To adjust temperature biases we employed the most accurate data available, i.e. measurements at the meteorological station, to correct the annual cycle of the seasonal forecast systems, to make it similar to the observed one. The adjustment of precipitation biases deserves more sophisticated techniques. Precipitation measurements in mountain areas are affected by large errors owing to wind drift and inadequacy of unheated and insufficiently-heated pluviometers, both leading to a large underestimation (Kochendorfer et al., 2017a, b). Clearly the
lack of reliable ground measurements hampers the possibility to accurately bias-adjust seasonal forecast precipitation data. In this study we adjusted precipitation forecasts with the quantile mapping method using ERA5 reanalysis as a reference data, assuming ERA5 to be an adequate approximation of the ground truth. Of course this is an assumption difficult to verify directly owing to the insufficiency of trusted in-situ data, while it is relatively easier to measure the effect of the precipitation adjustment on the simulated snow depth. Results show that the application of bias adjustment to original precipitation data
is useful in case of strong biases in the forecast system (MFS6), while it is less useful in case of small bias in the forecast system (ECMWFS5) and large uncertainties in the reference data. Surprisingly, lower RMSEs are obtained using the original seasonal forecasts of precipitation (RAW) rather than the bias corrected (QM) ones, and in both cases RMSE is lower than in the ERA5 experiment, suggesting poor agreement between ERA5 precipitation and the real one. This finding stresses the importance of having accurate precipitation observations to bias-adjust precipitation forecasts, and the need of more accurate
heated pluviometers to limit the snow undercatch. Despite the larger RMSE on the snow depth climatology, the difference in skill scores between RAW and QM experiment is generally very small. In fact, the scores of the QM experiment lie within the range of uncertainty of the score of the RAW experiment, so the bias adjustment does not substantially influence the skills of the prototype. These results are in agreement with a former study which found that the application of the quantile mapping to seasonal forecast products eliminates forecast biases in the reforecasts, without adding much to correlation skill (Becker,
Bernd Dieter, 2019).

## 4.3 The impact of the spatial downscaling of precipitation

The application of the RainFARM downscaling to precipitation seasonal forecasts has different effects on the model RMSE depending on the station but not on the forecast system considered. In fact, the successful application of the RainFARM method


(i.e. lower RMSE in the RF experiment compared to the RAW experiment) mainly depends on the accuracy of the reference
climatology used to derive the weights. If the reference climatology over- or under-estimates the impact of topography on local
precipitation amounts this feature will be reflected also in the downscaled data, irrespectively of the seasonal forecast system
employed. So, a locally inaccurate reference climatology introduces an additional source of error (see for example the case of
Lago Agnel station, RF vs. RAW experiments). Since the results are station-dependent, we recommend checking the effects of
the precipitation downscaling by verifying the improvement of the agreement between the simulated and the observed snow
depth climatologies. If results are not good one should consider either using another reference dataset with higher accuracy
or directly employing the original (RAW) precipitation at the coarse scale as input for the modeling chain. In support of this
last option are the results of the deterministic metrics, which do not show a significant increase in skill scores when using
downscaled data compared to original coarse scale data.

## 4.4 Spatial downscaling of other input variables

Apart from air temperature and precipitation the other variables necessary to drive the SNOWPACK model are critical to be
adjusted and/or downscaled mainly due to the lack of i) surface observations to be used as a reference for bias-adjustment and
ii) robust downscaling methods with proven effectiveness. Different methods have been developed to downscale wind fields,
based on cluster analysis (Mengelkamp et al., 1997; Salameh et al., 2009) or using a dynamical-statistical approach (Pryor
and Barthelmie, 2014), but all of these are affected by large uncertainties (Pryor and Hahmann, 2019). Martinez-García et al.
(2021) shows a comparison of different statistical methods, demonstrating the non-existence of an optimal approach for all
regions and applications. Humidity variables are rarely considered by downscaling studies. The most common approach con-
sists in usage of a stepwise multiple linear regression (Anandhi, 2011). The downscaling performance depends on predictors
selection, however upper air humidity variables are assessed as the most efficient ones. Regarding soil moisture, Peng et al.
(2017) performed a comparison of different downscaling approaches and highlighted limitations and uncertainties of these
methods, underlining the need of improving the accuracy of input remote-sensing data, and developing effective performance
metrics for the downscaling techniques. Spatial downscaling for incoming radiation is more complex than other variables. For
example, Gupta and Tarboton (2016) downscaled MERRA reanalysis data of incoming shortwave radiation by interpolating
them from coarse grid to DEM elevation one, while the incoming shortwave radiation is estimated from air temperature, cloud
cover and atmospheric emissivity. In that case, the downscaling did not reduce the uncertainty of raw data. Other approaches
exist, but they mostly treat global radiation (Fealy and Sweeney, 2008; Antonanzas-Torres et al., 2014). Since bias-adjustment
and downscaling techniques for variables other than temperature and precipitation are affected by large uncertainties, we pre-
ferred to i) verify the overall agreement between seasonal forecasts and corresponding station measurements (when available)
or ERA5 data over the period of study; provided acceptable agreement between the forecast and the reference dataset, ii) down-
scale seasonal forecasts using a simple bilinear interpolation to the coordinates of the station, which procedure is acceptable in
absence of more sophisticated methods.





## 4.5 Impact of the choice of the snow model

A variety of snow models with different degrees of complexity have been developed for different purposes and applications, from very simple empirical models (e.g. degree-day models) to sophisticated, multi-layer physical snow models. The choice of the appropriate complexity depends on the objective of the work. Förster et al. (2018) aims at forecasting February SWE anomalies spatially-averaged at the catchment scale, so they employed a simple hydrological snow model driven by air temperature and precipitation anomalies only, at coarse (monthly) time resolution. An advantage of this simple approach is the limited input data requirement and the low computational load of the simulations, at the expense of higher uncertainty in the output results. Our objective is to look with finer spatial detail, moving from the catchment scale to the local scale, and forecast monthly snow depth at specific sites of interest for economic activities. In this paper we adopted a sophisticated, physical, multi-layer snow model (SNOWPACK) which provides accurate daily snow depth estimates (RMSE= 0.10 m; BIAS=0.00, Pearson-Correlation=0.79 in NW Italian Alps) across a number of different conditions and seasons Terzago et al. (2020). The high level of accuracy of this model allows us to make the hypothesis that the model error is neglectable compared, for example, to the error associated with the forcing. This hypothesis simplifies the interpretation of the results and allows to better distinguish the contribution of the different elements of the modelling chain to the total error. The main drawback of using SNOWPACK is the number of input variables needed to run the simulations, that also limited the number of seasonal forecast systems that can be considered in this analysis.

## 4.6 Uncertainty in the validation data

The snow depth data used to evaluate snow forecasts are quality-controlled in-situ measurements, whose typical errors are on the order of a few centimeters. This approach allows to reduce the uncertainty associated the reference data compared to more common cases in which reference data are simulated by hydrological models and model errors affect the quality of the reference data (i.e. Förster et al., 2018).

## 4.7 Computational costs

The modelling framework presented in this study is quite complex and includes the following steps: i) download of ensemble seasonal forecast forcing; ii) bias adjustment of temperature and precipitation; iii) spatial downscaling (all variables); iv) temporal downscaling (all variables); v) SNOWPACK simulations; vi) post-processing of the SNOWACK forecasts; vii) generation of the plots; viii) update of the website. The most time-consuming steps are the bias-adjustment and the downscaling of the precipitation input. The bias-adjustment with the quantile mapping method can substantially improve the agreement between the modelled and the observed climatology, however it is found to have a small impact on the forecast skills, especially regarding tercile-based forecasts. The limited added value of precipitation bias adjustment and downscaling to the forecast skill seems to suggest that, in these sites and in these conditions, original RAW precipitation input can be employed obtaining similar results as in the more complex frameworks.





# 5 Conclusions

The paper presents first-of-their-kind multi-model ensemble seasonal forecasts of the snow depth evolution from November up to May of the following year (7 months lead time) and evaluates them at three study sites in the Italian Alps which are relevant

for water management, hydropower production and alpine ski tourism. The prototype to generate snow forecasts is based on the SNOWPACK model forced by meteorological data of two Copernicus Climate Data Store seasonal forecast systems, namely ECMWFS5 and MFS6. Forecast skill has been assessed employing both deterministic and probabilistic metrics, and using snow depth station measurements as a reference. The skill has been investigated also in relation to different levels of post-processing of total precipitation input, i.e. using raw, bias-corrected, downscaled, bias-corrected and downscaled precipitation

data, since this variable deeply affects snow dynamics and the goodness of snow simulations.

Many robust features have been found across different seasonal forecast systems, seasons, stations and scores. The prototype running from November $1^{st}$ up to 7 months lead time, shows surprisingly good skill at predicting the tercile category for different time aggregation of the snow forecasts: below- and above-normal winter (DJF), spring (MAM), and November-May average snow depth are predicted with higher accuracy (BSS) and higher discrimination (AUCSS) with respect to a simple

forecasting method based on the climatology. Ensemble mean monthly snow depth forecasts are significantly correlated with observations not only at short lead time 1 and 2 months (November and December) but also at lead time 5 and 6 months (March and April) when employing the ECMWFS5 forcing. Moreover the prototype shows skill at predicting extremely dry seasons, i.e. seasons with snow depth below the $10^{th}$ percentile, while the prediction of extremely wet seasons (i.e. snow depth above the $90^{th}$ percentile) is model-, station- and score-dependent. The bias adjustment of precipitation forecasts with the quantile

mapping technique can substantially improve the agreement between the modelled and the observed snow depth climatology provided that a reliable reference dataset is used. However, the application of bias-adjustment, downscaling or bias-adjustment and downscaling techniques does not result in remarkable differences on the skill scores compared to the case in which raw precipitation data are employed. This suggests that the probabilistic skill scores are weakly sensitive to the treatment of the precipitation input. The use of raw precipitation data allows simplifying the modelling chain and boosting the production of

snow forecasts at least at the three study sites considered. The exportability of these results to other study sites should be checked.

The predictability of the snowpack evolution at lead times up to 7 months is the major result of this study and corroborates the hypothesis that snowpack is a natural "integrator" of the climatic conditions (conditions of the meteorological drivers) at the monthly/seasonal scale, so even if the forecasts of the drivers (air temperature, precipitation, etc ... ) do not exactly match

the observations at sub-monthly time scales, the differences may compensate over monthly/seasonal time scales and provide reasonable monthly/seasonal snowpack forecasts. This is an important step forward in the seasonal prediction of hydrological variables: while the skill in streamflow prediction is limited, the storage of water within the snowpack can be predicted also at long lead time. This is particularly relevant in mountain catchments where most of the run-off in spring is due to snow melt, and the forecasts of below- or above-normal snow depth have immediate applications in the management of water resources,

hydropower production and ski resort management. A reliable seasonal forecasting system, e.g with a lead-time up to 3–6



months, could bring an important improvement in the long-term optimization of the energy production, since the hydropower reservoir management heavily depends on the expected seasonal hydrological characteristics, e.g. the snowpack development.

Although this prototype has been conceived to respond to practical needs of end users and it has been applied in specific study areas where forecasts were meaningful to them, it is extremely flexible and it can be applied to any other mountain areas,
provided that long-term temperature and snow depth time series are available for bias-correcting temperature forecasts and validating snow predictions, respectively.

In light of the exportability of this prototype to any mountain site, future work should be done to run this prototype at other sites of the Alps and beyond to further check its skill and to obtain a more complete picture of the snow forecasts for the season ahead along elevational transects or at the regional or even mountain range scale. These forecasts are particularly useful for all
activities and sectors related to snow-hydrological fields, i.e. for example irrigation consortia, industry, ski resort, hydropower plant and water resource managers. In addition, they help estimating the amount of water made available by snowmelt, mainly at the head of Alpine catchments, since in summer it accounts for almost the total runoff. This knowledge can help to better address problems related to dearth of water, as in periods of drought, or prevent disasters due to the opposite situation, such as floods in Alpine valleys.

*Data availability.* The datasets presented in this study can be obtained upon request to the corresponding author.

*Author contributions.* Original idea of the work: Jvh, ST; Development of the modelling chain: ST, with help of Jvh for bias-correction and downscaling tools; Run simulations: ST with help of GB; Data Analysis: GB; Writing first draft of the paper: ST and GB; Revision of the paper: Jvh and all

*Competing interests.* The authors declare that no competing interests are present.

*Acknowledgements.* This work was performed in the framework of the MEDSCOPE (MEDiterranean Services Chain based On climate PrEdictions) ERA4CS project (grant agreement no. 690462) funded by the European Union.



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
