# Peer review of "Seasonal forecasting of snow resources at Alpine sites"

_Hydrology and Earth System Sciences, 2022_

## Author Comment (AC1)

**Reply to Anonymous Referee #1**

Terzago et al. present some encouraging results on seasonal forecasting of snow at levels that could be of commercial use. There are overlaps with the aims of the PROSNOW project (http://prosnow.org/); I haven't seen a published demonstration of seasonal forecasting from that project, but Koberl et al. (2021) on the market for seasonal forecast services for ski resorts is relevant.

**Reply:** We thank the reviewer for this comment. Indeed there are common aims between this work and the PROSNOW project. The PROSNOW project developed a demonstrator of a meteorological-to-climate prediction and snow management system from a short term forecast covering the first 4 days to several months ahead, specifically tailored to the needs of the ski industry. Stimulated by the comment of the reviewer we have performed an indepth literature review on the papers related to this project. Ebner et al. (2021) evaluate the accuracy of the piste snow management module implemented within several snowpack models to simulate snow management for selected ski resorts. However no paper has been published on the seasonal prediction of snow resources. We agree that Koberl et al. is a relevant reference that illustrates the demand of climate services from the perspective of the ski resort managers. We have now added a reference to this paper in the Introduction.

**L 25: The conclusion that bias correction of precipitation forecasts has little influence on snow depth skill scores might be surprising, or it might be misleading as no observations are used in the bias correction of precipitation (which is not mentioned in the abstract).**

**Reply:** Thank you for this comment which allows us to better explain this point. The bias correction of precipitation forecasts is indeed essential to adjust biases eventually present in the seasonal forecast system outputs. For example, in our study we find that MFS6 largely overestimates total precipitation and this is reflected in disproportionately large snow depth values (Figure 5b of the manuscript): in such cases the application of bias correction methods is essential to reproduce realistic snow depth values and climatology. In particular, the bias correction which we applied to precipitation forecasts, despite being based on ERA5 rather than on observations, successfully adjusts the huge bias of the forecast snow depth climatology and makes it very close to the observed snow depth climatology (Figure 5b). Alternatively, if precipitation forecasts are already realistic, such as in the case of ECMWFS5 (Figure 5a), the application of a bias-correction has little impact on the snow depth climatology. So, in general, the application of bias correction methods to the forcing is necessary to take care of possible biases.

A different point is the influence of the bias correction of the skill of the forecast, and hence on the skill scores. Some skill scores are defined in such a way that they are not sensitive to the bias in the forecasts. The ROC curve, for example, is not sensitive to the bias in the forecasts, and a biased forecast may still have good resolution and produce a good ROC curve: in this case biased forecasts can be improved with bias correction/calibration and the metric gives an indication on the potential usefulness of the forecast. On the contrary, other skill scores such as the BSS and the CRPSS are sensitive to biases in the forecast distribution, so it is important to take care of and adjust biases. In conclusion, the application of bias adjustment methods to the main meteorological drivers of snow processes is needed to reconstruct a plausible snow depth climatology and a distribution as close as possible to the observed one, but it does not necessarily lead to improvements to skill scores. We have added the missing information on the bias correction with respect to ERA5 in the abstract and we have better clarified these points in the discussion of the manuscript.

**L70: The GEM15 forecasts used by Bellaire et al. (2011) were only out to 48 hours and were found to overestimate precipitation, so differences in conclusions from this study might be expected. Forster et al. (2018) is a much more direct precursor of this study.**

**Reply:** Thank you very much for the suggestion, indeed, since it is not totally relevant, we decided to remove this reference.

**L77: "this study" would more accurately be described as "that study", i.e. Forster et al. (2018), not Terzago et al. (2022).**

**Reply:** Modified as suggested by the reviewer, thank you.

**L109: As stakeholders were involved in designing the prototype system, it is curious that none are included in the author list or acknowledgements.**

**Reply:** Stakeholders are now mentioned in the acknowledgements, thank you for this comment.

**Table 1: 6 decimal places in latitude and longitude locate the stations to within 10 cm, which seems unnecessary.**

**Reply:** We agree that the information is very detailed, however we prefer to provide the
exact location of the station as indicated in the official database and website of the data
providersproviders(ARPAPiemonte).Thankyou.

**Table 1: "Total radiation" here is, I think, net radiation.**

**Reply:** "Total radiation" is the "total incoming shortwave radiation", we have specified it in the main text, thank you.

**Table 2 contains only a small amount of information that could easily be incorporated in the text.**

**Reply:** Information in Table 2 has been included in the main text (Section 2.3) and Table 2 has been removed.

Section 2.3.1: I find the bias correction hard to understand. What is the elevation of the ECMWFS5 temperature forecast in Figure 2a? Is it surprising that there is such a large *cold* bias compared with a station at 2410 m elevation? If the green line was produced by quantile mapping the raw data onto the ERA5 CDF, why is it further from ERA5 than the raw data? If the discontinuity in the green line is due to the monthly quantile mapping, why does it only appear in mid-February?

**Reply:** Thank you for this comment, we recognize that this section on the temperature bias correction was not clear and we have modified it in the revised version of the manuscript. Indeed analysis of the ECMWFS5 orography shows that the closest grid point is always at an elevation which is lower than that of the station by several hundreds of meters. Despite the lower elevation, ECMWFS5 shows a large and cold bias compared to observations at Bocchetta delle Pisse (see Figure 1a below) and smaller biases at the other stations, i.e. Rifugio Gastaldi and Lago Agnel (Figure 1b,c). This analysis confirms a systematic cold bias by the model which cannot be attributed simply to differences in elevation.

Figure 1 Near-surface air temperature climatology (1996-2016) for the ECMWFS5 seasonal forecast system, before (red) and after (green) the bias-adjustment, compared to surface station data, for the three stations considered, i.e. a) Bocchetta delle Pisse, b) Rifugio Gastaldi and c) Lago Agnel.

Regarding the temperature bias correction, in the submitted version of the manuscript we described two different methods: a more complex one based on the quantile mapping of the temperature forecasts to ERA5, and a simpler one based on the adjustment of the average bias with respect to observations. The first method, based on ERA5, is indeed not optimal mainly because ERA5 is as well biased compared to observations (Figure 2a of the manuscript) and thus not suitable as a reference dataset for the bias correction of temperature. This issue is evident also before applying the quantile mapping (Figure 2a), so the discussion on the application of this method using ERA5 as a reference is indeed not useful to the reader; moreover, Figure 2a contains an error in the quantile mapped data, as promptly noted by the reviewer. In fact, the curves corresponding to ECMWFS5 after the quantile mapping (green) and after the downscaling to the station (light blue) are not consistent with ERA5 temperatures, as both lines are further from ERA5 than the original raw data. Given this issue, and given that this method was discarded and not used further in the analysis, we prefer to remove it in the revised version of the manuscript and to keep only the description of the bias correction method based on the adjustment of 6-hourly average bias with respect to temperature observations, which is effective in reconstructing the observed climatology at all the sites considered (Figure 1) and has been used in the manuscript.

**Table 3: Use superscripts for W/m2**

**Reply:** Done, thank you.

**Figure 2: The x-axis is labelled in months, not days. "Downscaled data" means different things for temperature and precipitation that is not apparent from the figure or caption. Why does cumulated precipitation appear to decrease in mid-February?**

**Reply**: We have modified Figure 2 according to the reviewer's suggestion. In the new version of Figure 2 the label of the x-axis has been removed. The caption has been modified to include an explanation of the differences among the temperature and the precipitation downscaling. Regarding the discontinuity in the precipitation climatology, it occurs in correspondence of February 29th, when the multi-year average is calculated over the leap years only instead of the full period. In the new plot we removed the data corresponding to the 29th February for clarity.

**L264: "(" missing before Matheson.**

**Reply:** We have corrected this, thank you.

**L276: subscript perf**

**Reply**: We have corrected this, thank you.

**Figure 3: The blue and dark blue lines are hard to distinguish when printed.**

**Reply:** We have changed dark blue lines into black in the final figure.

---

## Author Comment (AC2)

**Reply to Anonymous Referee #2**
**The discussion paper "Seasonal forecasting of snow resources at Alpine sites" by Terzago et al. comprises a thorough analysis of the capabilities of a new framework to predict snow depths over lead times covering the entire winter season. To this end, the authors ran the physically based snowmodel SNOWPACK using input data from seasonal weather forecasting systems by ECMWF and Météo France that were downscaled and debiased to the location of three stations in the Western Italian Alps. The analysis shows some prognostic skill of the framework relative to a climatology-based reference guess.**

**While an interesting topic and a well-prepared manuscript, my main concern arises from the fact that no (trusted) local precipitation measurements were available. This necessitated the use of ERA5 data as ground truth or target, which likely derailed some of the findings, in particular those that deal with the effect of downscaling and debiasing. I acknowledge that not having accurate precipitation data is the norm, in particular in such a use-case driven study. Yet, the absence of local precipitation data is critical to a point, where I wonder how useful certain sections of this study are at all (4.2 – 4.4). In this situation, I recommend to infer local (solid) precipitation from snow depth using data assimilation and use this in lieu of measurements. This approach would be in line with the authors' underlying premise that the snow model is the strongest members in their system (see line 582). I think SNOWPACK even has an in-built data assimilation mode that allows to use snow depth instead of precipitation as input.**

**Reply:** We acknowledge that the use of ERA reanalysis as a reference for the bias-correction of precipitation can lead to uncertainties in the bias-corrected data. For this reason, following the suggestion of the reviewer, we have inferred total precipitation from snow depth records using the parameterizations already available in the SNOWPACK model. The assimilation of snow depth data in the SNOWPACK model allowed us to obtain a new daily total precipitation time series, which has been compared to the ERA5 total precipitation data in terms of seasonal cycle and probability density function (PDF) of daily precipitation. The results are shown in Figure 2.

[Figure]

Figure 2: Comparison between ERA5 (black) and SNOWPACK-derived (red) total precipitation: a) seasonal cycle of total precipitation from November to May expressed in terms of multiannual mean and corresponding standard deviation, and b) probability density function (PDF) of daily precipitation in the two datasets.

Figure 2a shows the seasonal cycle of total precipitation from November to May in the two datasets, and in particular the multi-annual mean and the standard deviation of monthly values, the latter being an indicator of interannual variability. ERA5 and SNOWPACK seasonal cycles are quite similar, with ERA5 laying in the range of the SNOWPACK mean ±1 standard deviation. Looking at the differences between the two datasets, ERA5 shows slightly higher monthly means in late winter (JFM) and late spring (May), and less interannual variability (smaller standard deviation) compared to SNOWPACK. More in detail, SNOWPACK allows for lower monthly precipitation values and lower daily precipitation values in the range between few mm/day and 20 mm/day (Figure 2b), while above this threshold the PDFs of the two datasets are similar. In conclusion, although with these differences, both ERA5 and SNOWPACK total precipitation datasets can be considered valid approximations of the observed precipitation amount, in absence of more accurate observational data. From this analysis we do not expect that the use of ERA5 as a reference for bias correction introduces large discrepancies in the bias corrected data. This is also confirmed by the fact that precipitation bias correction successfully adjusts the huge bias of the MFS6 snow depth climatology and makes it very close to the observed snow depth climatology (Figure 5b of the paper). We discuss this topic more in detail in Section 4.2 on the impact of the bias correction.

**Another concern is the lack of discussion about the strength of the forecasting skills. In particular for readers who do not juggle with BBS, CRPSS, or AUCSS on a daily basis, wording such as "demonstrates skill", "show skill", or "surprisingly good skill" is not really meaningful. Is "skill" just slightly better than guessing? The authors have quantitative data, but need to put them in context. I found an interesting statement in line 490: "seasonal forecasts of snow depth appear more robust than streamflow forecasts". This is where quantitative reference should be made to corresponding score data from the evaluation of seasonal streamflow forecasts.**

**Reply:** The evaluation of the forecast characteristics through the various scores and skill scores is a standard in forecast verification, even though not immediately understandable

to the non-expert readers. Further explanation on these metrics can be found for example in Calì Quaglia et al., 2021. To help the reader, in our manuscript we employ skill scores only. Skill scores measure the improvement (or the worsening) of a given forecast method compared to a trivial forecast based on the climatology, the persistence of the observed anomaly, etc. The more positive is the skill score, the better is the quality of the forecasts; the more negative is the skill score, the worse is the forecast quality and hence the quality of the forecast method. The added value of the new forecast method is directly understandable from the sign and the absolute value of the skill score. Regarding the statement mentioned by the reviewer, i.e. seasonal forecasts of snow depth seems to be more robust than streamflow forecast: the statement is probably intuitive but should be taken with caution since it is only qualitative and not quantitative. Statements based on skill scores are instead quantitative and more rigorous, so they are the preferred way to convey the results of our analysis. To facilitate the understanding we revised the text of the manuscript improving the explanation of how to interpret skill scores.

Calì Quaglia, F., Terzago, S., and von Hardenberg, J.: Temperature and precipitation seasonal forecasts over the Mediterranean region: added value compared to simple forecasting methods, Climate Dynamics, pp. 1–25, 2021

**Further specific comments:**

**Line 86 / 603: In the context of this study "multi-model" arguably suggests that there is more than one snow model involved, which is not the case. Please revise.**

**Reply:** "Multi-model" has been changed with "multi-system" throughout the text.

**Section 2.3.3: Was local terrain accounted for in the downscaling (sky view for longwave, and terrain shading for shortwave)?**

**Reply**: We did not take into account local sky view and terrain shading in the downscaling of radiation.  We performed a basic linear interpolation to the station coordinates without considering local factors. This simple approach has been better clarified in the text in Section 2.3.3.

**Section 2.4: Even if the authors use soil temperature boundary conditions, I would recommend a spin-up to allow for a realistic initial soil temperature profile for all simulations, i.e. also if snow depth was zero on Nov-1.**

**Reply**: In the SNOWPACK model set-up which we adopted, we do not need to provide the initial soil temperature profile to the model. All SNOWPACK simulations are performed providing the ground temperature, i.e. the soil temperature in the topmost part of the soil at the snow–soil interface, as an input variable. The ground temperature is the unique soil "boundary condition", and we assume that deep soil layers do not affect the snowpack dynamics, so no soil layer is considered. This assumption is based on the fact that in all the three sites considered the soil is usually continuously covered by snow from early November to the end of the simulation period in May (see Figure 3 below with the

climatological mean values). In general, we can say that from November to May the ground is continuously covered by snow and that i) the ground temperature remains close to 0°C during that period, ii) the soil temperature at lower depths does not fall well below 0°C (i.e. no significant soil freezing occurs during the simulation period) due to the isolating effect of the overlying snowpack. This condition is documented for example in Wever et al., 2015 (Figure 7 of that manuscript) for the Alpine site of Weissfluhjoch, 2540 m a.s.l., located at similar elevation as the stations considered in this study for which no soil temperature measurements are available.

Given this SNOWPACK model setup, we do not need to provide the initial conditions of the soil layers, so we do not need to perform the spin-up to generate them. The spin-up is performed only when snow depth is already present at the beginning of the forecast period (i.e. November 1st): in this case we perform the model spin-up to reconstruct the snow profile, in terms of number of snow layers, layers' thickness, temperature, ice/liquid water content, snow density, etc. This has been better explained in Section 2.4.

[Figure]

Figure 3: Observed monthly snow depth climatology over the period 1995-2015 for the three stations considered in this study

**Section 4.5: I generally appreciate the approach of using a complex and trusted snow model to focus the uncertainty analysis to the forcing data and the models / methods used to derive them. However, at the same time I wonder, if a temperature index model would actually provide better results, because a) it only uses two input parameters and avoids deteriorated performance due to uncertainties associated with the other forcing fields (wind, radiation, …); b) local calibration of a temperature index model is simple and fast, and can compensate for remaining systematic biases, e.g. arising from the lack of local precipitation data. It's probably not realistic to expect the authors to perform such a comparison as part of this paper, yet, the above consideration would make a useful complement to section 4.5.**

**Reply:** We thank the reviewer for this comment. When planning the experiments we carefully discussed what type of snow model was best for our purpose. We considered a range of models, from simple degree-day models to the most sophisticated ones such as SNOWPACK. Simple models have the advantage of requiring few input parameters, so they allow to avoid uncertainties associated with other forcing fields, as said by the reviewer. The disadvantage of these models is that they need to be calibrated over each study site, so sufficiently long time series of forcing and validation data are necessary to calibrate and

validate the model over independent time periods. However, retrospective seasonal forecasts cover a period of 20-25 years, which is already quite short, and further dividing it into two parts would lead to unrepresentative results. On the other hand, sophisticated snow models have higher input requirements and higher computation-load compared to simple snow models, but they have the advantage that they can be directly used without calibration and their snow snow estimates have high accuracy (Terzago et al., 2020) so one can make the hypothesis that the model uncertainty is neglectable with respect to the uncertainty on the forcing. After considering all these points, we agreed that it is better to employ a sophisticated snow model, which guarantees higher accuracy and lower model uncertainty compared to a degree-day model. We better explained this in Section 4.5

Terzago, S., Andreoli, V., Arduini, G., Balsamo, G., Campo, L., Cassardo, C., Cremonese, E., Dolia, D., Gabellani, S., von Hardenberg, J., Morra di Cella, U., Palazzi, E., Piazzi, G., Pogliotti, P., and Provenzale, A.: Sensitivity of snow models to the accuracy of meteorological forcings in mountain environments, Hydrol. Earth Syst. Sci., 24, 4061–4090, https://doi.org/10.5194/hess-24-4061-2020, 2020.

**Line 627: Being able to predict that snow will melt at a certain location in, say, May is fairly easy even if the forecasted weather data is uncertain. Having said that, the increase of some of the skill scores towards the end of the season is no real surprise.**

**Reply:** Snow depth seasonal forecasts do not provide only the information mentioned by the Reviewer, i.e. that snow will melt at the end of the snow season, which would be a quite obvious result. They also provide information on the expected snow depth anomaly with respect to the average conditions for the period. In other words, the forecast answers the questions: what is the deviation (excess or shortage) of the forecasted snow depth from the average snow depth value in May? What is the probability that in May snow depth will be below/near/above normal conditions for that period? Seasonal forecasts predict the expected distance of a given variable from the normal ("average") conditions. So the skill in correctly predicting the anomaly at long lead times is indeed quite surprising and a non-obvious result. The snowpack predictability that we find in late spring, i.e. April, can be probably explained by the fact that April is the snow depth accumulation is maximum (see Figure 3) and snowpack is an integrator of the weather conditions and snowfalls over the previous snow season up to April, so even if seasonal forecast models are not able to capture the correct timing/amount of snowfalls, they seem to be able to predict the overall conditions in the winter/early spring period. This has been better clarified in the discussion, thank you for the comment.

---

## Author Response (AR2)

Dear Editor,

please find enclosed the revised manuscript which has been modified according to the comments of Reviewer#2, i.e:

**Reviewer2: "Results and discussion only report whether or not AUCSS, CRPS, or BSS are above zero, but there seems to be no attempt to report or discuss why/when/where a score is 0.2, 0.4, or 0.6, which IMO is a missed opportunity. Any additional use of these 'hues of green/red' would be appreciated."**

Reply: In order to meet this request, in the new version of the manuscript the plots showing BSS, AUCSS, CRPSS (Figure 8, 10 and 12) have been updated. In the new plots we used a discretised scale with thresholds of 0, ±0.2, ±0.4 and we introduced a new classification of the forecast skill, distinguishing between "fair", "good", and "remarkable" skill: "fair" corresponds to skill score (SS) $0<SS\leq0.2$, "good" corresponds to $0.2<SS\leq0.4$, and "remarkable" corresponds to $SS>0.4$). This approach allowed to extract the information requested by the reviewer concerning "when" and "where" skill scores lie in those ranges; we could not discuss "why" the skill scores lie in those ranges since it would have required extensive work and new analysis, and this would be probably more appropriate for a separate paper. The information inferred from the new plots has been added in the manuscript in the corresponding sections. Lastly, concerning the new plots, we changed the color palette in order to meet the Color Blindness requirements.

**Reviewer2: "Please correct 'adjustment' in Line 549"**
Reply: Done

We hope that the revised version of the manuscript will be positively considered.

With kind regards
Silvia Terzago and coauthors